# Distinct mechanisms regulate ventricular and atrial chamber wall formation

Marga Albu [1,2,3], Eileen Affolter[1,2,3], Alessandra Gentile [1,2,3,6], Yanli Xu[1,2,3], Khrievono Kikhi [2,3,4], Sarah Howard [1,2,3], Carsten Kuenne [5], Rashmi Priya[1,2,3,7], Felix Gunawan [1,2,3,8] & Didier Y. R. Stainier [1,2,3] ✉

Tissues undergo distinct morphogenetic processes to achieve similarly shaped structures. In the heart, cardiomyocytes in both the ventricle and atrium build internal structures for efficient contraction. Ventricular wall formation (trabeculation) is initiated by cardiomyocyte delamination. How cardiomyocytes build the atrial wall is poorly understood. Using longitudinal imaging in zebrafish, we found that at least 25% of the atrial cardiomyocytes elongate along the long axis of the heart. These cell shape changes result in cell intercalation and convergent thickening, leading to the formation of the internal muscle network. We tested factors important for ventricular trabeculation including Nrg/ErbB and Notch signaling and found no evidence for their role in atrial muscle network formation. Instead, our data suggest that atrial cardiomyocyte elongation is regulated by Yap, which has not been implicated in trabeculation. Altogether, these data indicate that distinct cellular and molecular mechanisms build the internal muscle structures in the atrium and ventricle.

The vertebrate heart consists of distinct chambers: the atria receive the blood from the circulation whereas the ventricles pump blood out of the heart into the circulation. Both the atrial and ventricular muscle walls are composed of complex inner structures important for the function of each chamber. However, it remains unclear whether these internal muscle structures in the different chambers are built through similar or distinct processes.

Cardiac ventricles form complex inner muscular networks known as trabeculae, which increase tissue mass and contraction force, thereby playing critical roles in cardiac function[1,2]. Cardiac atria form a similarly complex muscular network wherein a large crista terminalis (terminal crest) splits into thinner parallel pectinate muscles and Bachmann's bundle. Together, these structures play important roles in action potential propagation and chamber contraction[1–9]. Since cardiac malformations are the most prevalent form of congenital defects[10], how the embryonic heart develops to achieve its architecture has been under intense investigation. In addition, abnormal atrial architecture can lead to serious symptoms including arrhythmias[3,7,8,11,12].

Atrial and ventricular cardiomyocytes (CMs) are known to have different cellular and molecular profiles as they differentiate[13–17]. In zebrafish embryos, atrial CMs initially appear more squamous while ventricular CMs appear more cuboidal[15,18,19]. At the molecular level, atrial and ventricular CM specification and differentiation are regulated in part by different pathways[15]. For example, the BMP pathway promotes atrial CM specification[19,20] whereas the FGF pathway promotes ventricular CM specification[21–23]. In addition, atrial and ventricular CMs cluster separately in scRNA-sequencing datasets[14,24–27], as they express different genes such as *MYH6* in atrial CMs and *MYH7* in ventricular CMs in fish and mammals[15,25,28–30]. Nevertheless, how these cellular and molecular differences affect morphogenetic mechanisms remains unexplored.

In the ventricle, the onset of trabeculation has been extensively investigated[2,31–53]. Trabeculae appear when a subset of the ventricular

[1]Max Planck Institute for Heart and Lung Research, Department of Developmental Genetics, Bad Nauheim, Germany. [2]German Centre for Cardiovascular Research (DZHK), Partner Site Rhine-Main, Bad Nauheim, Germany. [3]Cardio-Pulmonary Institute (CPI), Bad Nauheim, Germany. [4]Flow Cytometry Service Group, Max Planck for Heart and Lung Research, Bad Nauheim, Germany. [5]Bioinformatics Core Unit (BCU), Max Planck Institute for Heart and Lung Research, Bad Nauheim, Germany. [6]Present address: MRC Centre for Neurodevelopmental Disorders, King's College, London, UK. [7]Present address: Francis Crick Institute, London, UK. [8]Present address: Institute of Cell Biology, University of Münster, Münster, Germany. ✉e-mail: didier.stainier@mpi-bn.mpg.de

CMs, which initially form a polarized epithelium, delaminate to form multicellular clusters in the cardiac lumen. This delamination process requires Nrg/ErbB and Notch signaling[2,32–34,40–49,52,53]. Furthermore, blood flow/cardiac contractility is required for ventricular trabeculation, at least in part by promoting *nrg2a* expression in the endocardium[46,54,55].

However, the cellular and molecular factors involved in atrial wall formation are unclear. Studies in chick have shown that as the atrium develops, inner wall CMs become distinct from outer wall CMs by expressing genes involved in action potential generation and propagation such as *Nav1.4* and *Cx40*, and by becoming more proliferative[7,56]. Furthermore, a clonal study in zebrafish suggests that up to 7 days post fertilization (dpf), atrial CMs display a squamous and round morphology, whereas at 14 dpf, the atrium appears to consist of a webbed network of rod shaped CMs[57]. However, the exact mechanisms involved in atrial morphogenesis remain unclear. Here, we use the zebrafish model to investigate this process as it allows for high-resolution longitudinal imaging[51].

## Results

### A subset of atrial cardiomyocytes elongates in the direction of blood flow

In zebrafish, a contracting heart tube is visible by 24 h post fertilization (hpf)[58], after which the heart loops to form the atrium and ventricle, which are separated by the atrioventricular canal (AVC)[2]. The heart then undergoes several morphogenetic events including AV valve formation starting around 56 hpf[59], ventricular trabeculation starting around 60 hpf[49], and atrial chamber morphogenesis[57].

To determine the onset of atrial morphogenesis, we longitudinally imaged atrial development, and documented changes in cell behaviors that might lead to internal muscle structure formation. Taking advantage of the Airyscan imaging modality (Fig. 1a), we obtained clear 3D images of the atrium (Fig. 1b, b'; Supplementary Movie 1). To investigate atrial CM behavior, we imaged zebrafish larvae every 24 h, from 100 to 148 hpf, and observed a subset of CMs gradually elongating over this period (Fig. 1c–e'). To determine when cell elongation begins, we segmented atrial CMs to obtain their 3D shape index – a parameter describing cell shape in 3D - which revealed that a few atrial CMs start to elongate at 76 hpf, and that these shape changes become more prominent with time (Fig. 1f, g); at 100 hpf, approximately 15% of the atrial CMs are elongating, and at 124 hpf approximately 25% (Fig. 1g). During this cell elongation process, atrial CMs orient bidirectionally along the long axis of the heart (Fig. 1b', h–h'''). These data show that atrial CMs undergo a cell behavior that is distinct from those taking place during ventricular trabeculation, i.e., CM apical constriction and delamination[51], or from the previously hypothesized mechanism that drives atrial morphogenesis, i.e., division, budding and branching[57].

### Atrial cardiomyocyte elongation leads to cell intercalation

We hypothesized that the observed atrial CM elongation drives tissue architecture changes important for internal muscle structure formation. To test this hypothesis, we imaged larval hearts that mosaically express a cytoplasmic marker in atrial CMs, an approach that enables reliable tracking of individual cells (Fig. 2a–d). We then segmented neighboring and elongating CMs in 3D, prior to and during cell elongation. To improve visualization, we created both opaque (Fig. 2a'–d') and transparent (Fig. 2a''–d'') 3D surfaces. Through this method, we observed that cell elongation leads to cell intercalation, as revealed by the overlap of the mTagBFP⁻ (white surfaces) and mTagBFP⁺ (magenta surfaces) atrial CMs (Fig. 2a–d''). To understand how these beating CMs remain attached while intercalating, we examined the localization of the CM-specific cadherin, N-cadherin, in the region of cell intercalation (Fig. 2a'''–d'''). As atrial CMs elongate and intercalate, N-cadherin relocalizes from a continuous distribution on their lateral

membranes to a punctate distribution on their apical and basal membranes (Fig. 2a'''–d'''; Supplementary Fig. 1a–b'') similar to what was observed during ventricular CM delamination[60]. We also noticed that while elongating, CMs form apicobasal N-cadherin-based adhesions, whereas the CMs that maintain their round configuration maintain their lateral N-cadherin based adhesions (Supplementary Fig. 1a–g). In addition, unlike compact layer ventricular CMs, some atrial CMs detached from their neighbors, as revealed by an actin ring (Supplementary Fig. 1h, h') devoid of N-cadherin (Supplementary Fig. 1i, i'), possibly corresponding to the previously described regions of myocardial absence in the mature atrium[57]. In summary, we observed that atrial CM elongation leads to regional multilayering via cell intercalation, accompanied by changes in intercellular adhesion.

### Convergent thickening leads to internal muscle structures in the atrium

We then asked whether this cell intercalation-driven multilayering, also known as convergent thickening[61], builds the muscle structures inside the atrial wall. We created 3D surfaces of the atrial myocardium before (Fig. 2e, e') and after (Fig. 2f, f') CM elongation and observed ridges on the inner surface of 7 dpf atria. We thus hypothesized that the generation of these ridge-like structures in the atrium constitutes the first step in the formation of internal muscle structures, which are clearly present at 14 dpf (Fig. 3a, a')[57].

To test the hypothesis that the elongating CMs build these complex atrial muscle structures, we first segmented them from the rest of the myocardium at 14 dpf (Fig. 3a–c; Supplementary Movie 2). We observed that the inner muscle structures of the atrium were entirely composed of elongated CMs (Fig. 3a'–c'). We then used mosaic labeling to track the same atrial CMs from 124 hpf (Supplementary Fig. 2a–c) to 14 dpf (Supplementary Fig. 2a'–c') and observed that elongating CMs present at 124 hpf form the internal muscle structures at 14 dpf. Furthermore, using this mosaic labeling in combination with longitudinal imaging, we determined and compared the shape index of mTagBFP⁺ atrial CMs at 124 hpf and 14 dpf. Through this analysis, we observed that a subset of round atrial CMs had become elongated by 14 dpf, leading to a high F-ratio value (Supplementary Fig. 2d). However, elongated atrial CMs never reverted to the round configuration and maintained their shape (Supplementary Fig. 2e). And while all the CMs in the inner layer of 14 dpf atria were elongated, the outer layer contained both elongated and round CMs (Supplementary Fig. 2f). Altogether, these data indicate that CM elongation and intercalation build the internal muscle structures in the zebrafish atrium.

We then investigated whether atrial CM elongation correlates with changes in myofibril organization. We noticed that at 14 dpf, myofibrils of the inner atrial muscle network, which is composed of elongated CMs, are all oriented in the same direction, in contrast to the myofibrils of the outer layer CMs, which display a stochastic arrangement (Supplementary Fig. 3a–c'). We also tracked the myofibrils of the elongating atrial CMs and observed that prior to cell elongation, they display a stochastic orientation (Supplementary Fig. 3d, d'). However, during CM elongation, these myofibrils start aligning in the direction of cell elongation and become significantly thicker (Supplementary Fig. 3d–g). Altogether, these data suggest that atrial CM elongation might regulate myofibril organization, and thus possibly cardiac contraction. Therefore, we hypothesized that atrial CM elongation helps increase contraction forces. We tested this hypothesis by first measuring the atrial ejection fraction before (74 hfp) and after (124 hpf) CM elongation (Supplementary Fig. 3h–i') and found a significant increase during this time (Supplementary Fig. 3j). Altogether, these data suggest that atrial CM elongation, and associated myofibril maturation, are important to increase atrial contraction forces.

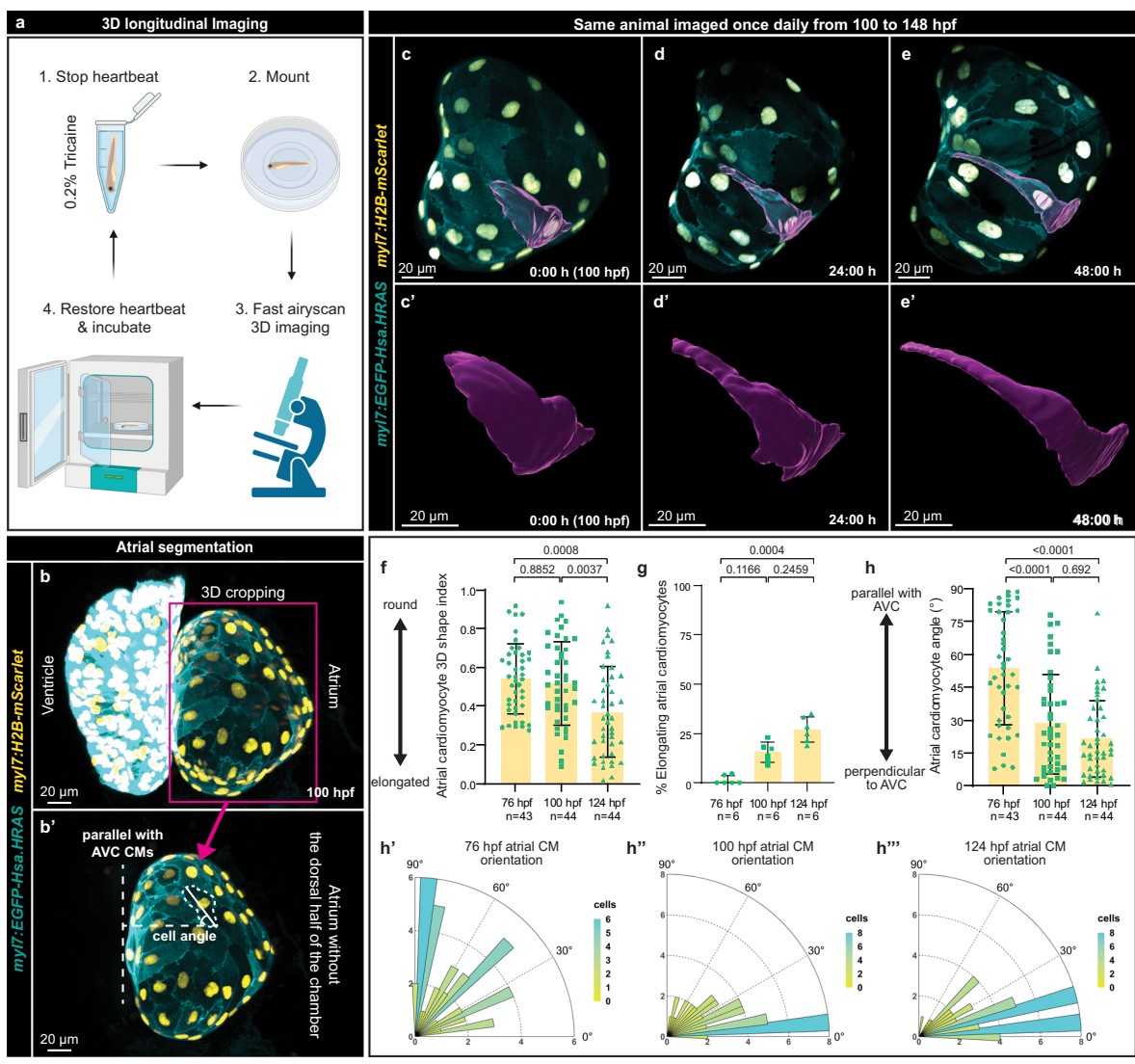

**Fig. 1 | A subset of atrial cardiomyocytes gradually elongate along the long axis of the heart. a** Schematic of 3D longitudinal imaging. Created with BioRender.com, released under a Creative Commons Attribution-NonCommercial-NoDerivs 4.0 International license. **b, b'** 3D image processing used throughout the study. 3D airyscan images of a 100 hpf larva showing both the atrium and ventricle (**b**) and only the atrium (**b'**) after 3D cropping in Imaris; CM membranes shown in cyan (*myl7:EGFP-Hsa.HRAS*) and CM nuclei in yellow (*myl7:H2B-mScarlet*); magenta square and arrow indicate cropped region; the solid line in b' is the longest line through one atrial CM, and the dashed lines show how the angle (relative to the AVC) of the highlighted atrial CM (yellow dashed outline) was measured. **c–e'** 3D airyscan longitudinal imaging of the atrium at 100, 124, and 146 hpf; CM membranes shown in cyan (*myl7:EGFP-Hsa.HRAS*) and CM nuclei in yellow (*myl7:H2B-*

*mScarlet*); one elongating CM shown in 3D (magenta). **f** 3D quantification of atrial CM shape at 76, 100, and 124 hpf (*n* = 43 atrial CMs from 6 hearts at 76 hpf, and n = 44 atrial CMs from 6 hearts at 100 and 124 hpf; each data point represents one atrial CM; average of 8 CMs per heart; ordinary one-way ANOVA with Tukey's multiple comparison test). **g** Percentage of elongating atrial CMs at 76, 100, and 124 hpf (*n* = 6 hearts for all time points; each data point represents one heart; average of 20 CMs per heart; Kruskal-Wallis test with Dunn's multiple comparison test). **h–h''**, Atrial CM angle measured at 76, 100, and 124 hpf (*n* = 43 atrial CMs from 6 hearts at 76 hpf, and *n* = 44 atrial CMs from 6 hearts at 100 and 124 hpf; each data point represents one atrial CM; average of 8 CMs per heart; Kruskal-Wallis test with Dunn's multiple comparison test; 76 vs 100 hpf *p*-value = 0.32*$10^{-4}$, 76 vs 124 hpf *p*-value = 0.67*$10^{-7}$). Error bars are mean ± SD.

## Membrane protrusions drive atrial cardiomyocyte elongation

We next aimed to uncover the cellular processes that underlie atrial CM elongation. High-resolution imaging of the atrial CM membranes prior to cell elongation revealed small protrusions (Fig. 2e), leading us to hypothesize that atrial elongation is an active process driven by membrane protrusion formation. To test this hypothesis, we labeled the membrane protrusions of a few CMs through the mosaic over-expression of PH-Akt1-tdTomato-PEST, a tagged membrane-binding domain of zebrafish Akt1 (Fig. 4a). [AKT1 is known to localize to the leading front of migrating cells[62,63], thereby enabling better visualization of membrane protrusions]. We tracked PH-Akt1-tdTomato-PEST+ atrial CMs and observed again their bidirectional elongation and the formation of membrane protrusions that appeared similar to

lamellipodia and filopodia (Fig. 4b–d')[64–66]. Lamellipodia are large membrane protrusions conventionally thought to be the primary drivers of cell migration, whereas filopodia are smaller protrusions that sense the environment and direct movement[64–66]. To test whether membrane protrusion formation drives atrial CM elongation, we overexpressed a dominant negative (DN) version of IRSp53 in CMs by using a *Tg(UAS:IRSp53DN-RFP)* line[67] crossed to the *Tg(myl7:GAL4)* line[68]. IRSp53[DN] inhibits actin filament formation in protrusions[67,69,70], and upon CM specific IRSp53[DN] overexpression, the proportion of elongating atrial CMs (Fig. 4e–g) and their orientation (Fig. 4e, f, h–h'') were significantly affected. In addition, we pharmacologically inhibited Rac1, a GTPase required for lamellipodia formation, and observed a significant reduction in atrial CM elongation (Fig. 4i–k), but not

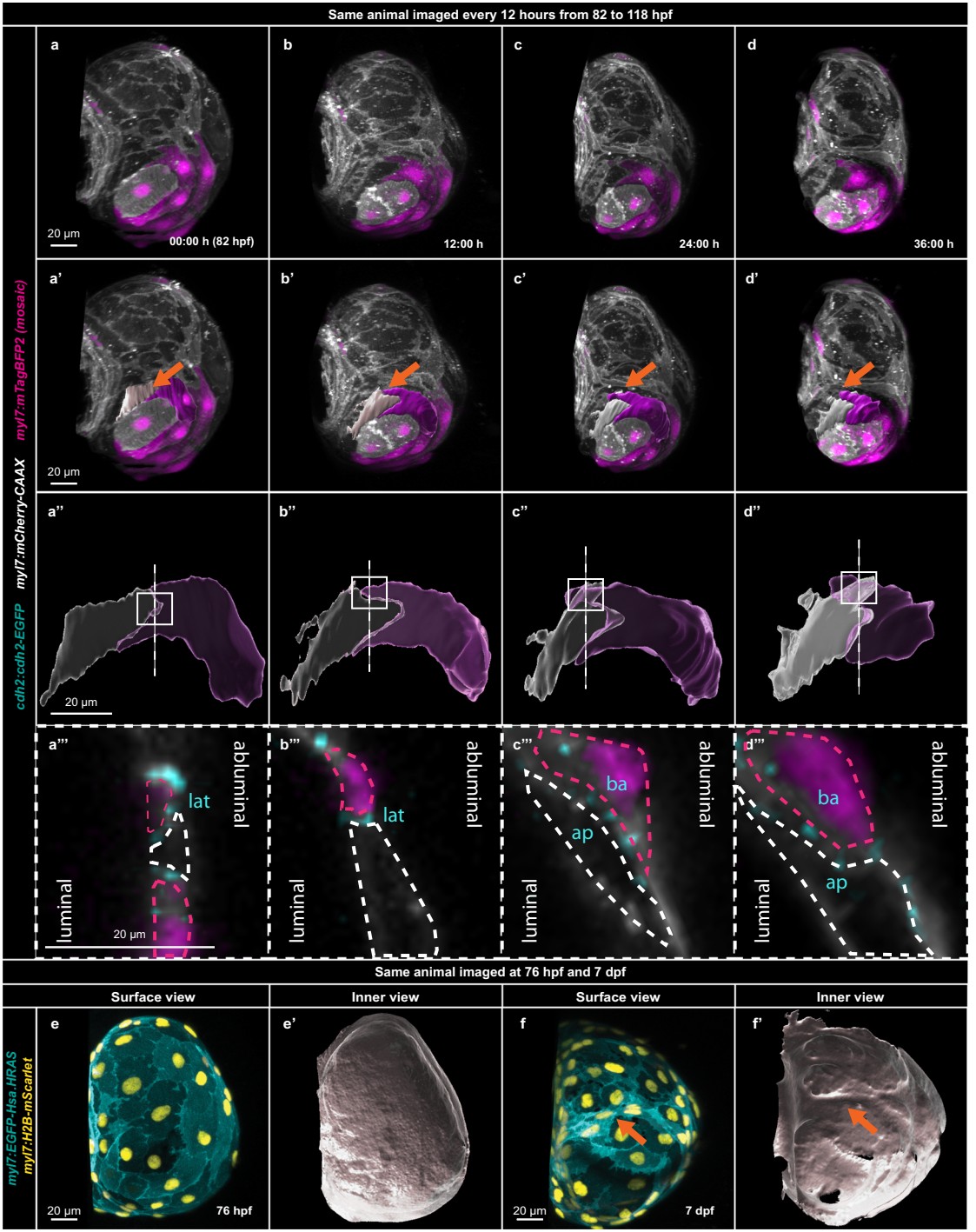

**Fig. 2 | Atrial cardiomyocyte elongation leads to cell intercalation and convergent thickening. a–d** 3D airyscan imaging of the same larva every 12 h from 82 to 118 hpf; CM membranes shown in white (*myl7:mCherry-CAAX*) and mosaic CM cytoplasmic expression in magenta (*myl7:mTagBFP2*). **a′–d′** 3D segmentation of two elongating and intercalating CMs reconstructed with opaque (**a′–d′**) and transparent (**a″–d″**) surfaces, revealing cell intercalation; orange arrow points to the segmented CMs; squares and dashed lines indicate cross-section region. **a‴–d‴**, Cross-sections through elongating atrial CMs; dashed lines outline the two intercalating CMs; N-cadherin shown in cyan (*cdh2:cdh2-EGFP*); lat–lateral adhesion; ap–apical adhesion, ba – basal adhesion; (**a–d″**) all 3D surfaces of the mTagBFP2+ CM shown in magenta and all 3D surfaces of the mTagBFP2- CM in white. **e, f** Outer surface views of 3D airyscan images of the same atrium at 76 hpf and 7 dpf; CM membranes shown in cyan (*myl7:EGFP-Hsa.HRAS*) and CM nuclei in yellow (*myl7:H2B-mScarlet*). **e′, f′** Inner surface views of the same 3D airyscan images at 76 hpf and 7 dpf; orange arrows point to CMs in the inner ridges.

orientation (Fig. 4i, j, l–l″). These data are consistent with the role of lamellipodia in driving rather than directing cell migration, contrary to filopodia.

Membrane protrusion formation and stabilization are regulated by cell contractility, which depends upon non-muscle myosin activity[66]. To test the role of non-muscle myosin activity in CM elongation, we overexpressed constitutively active (CA) and DN forms of the non-muscle myosin MYL9 in CMs using the *myl7* promoter. Overexpressing MYL9$^{CA}$ significantly increased atrial CM elongation, whereas overexpressing MYL9$^{DN}$ reduced it (Supplementary Fig. 4a–d). As *myl7* is active from 15 hpf[71], we tested whether atrial CM elongation could be induced to start prematurely or whether there is a mechanism

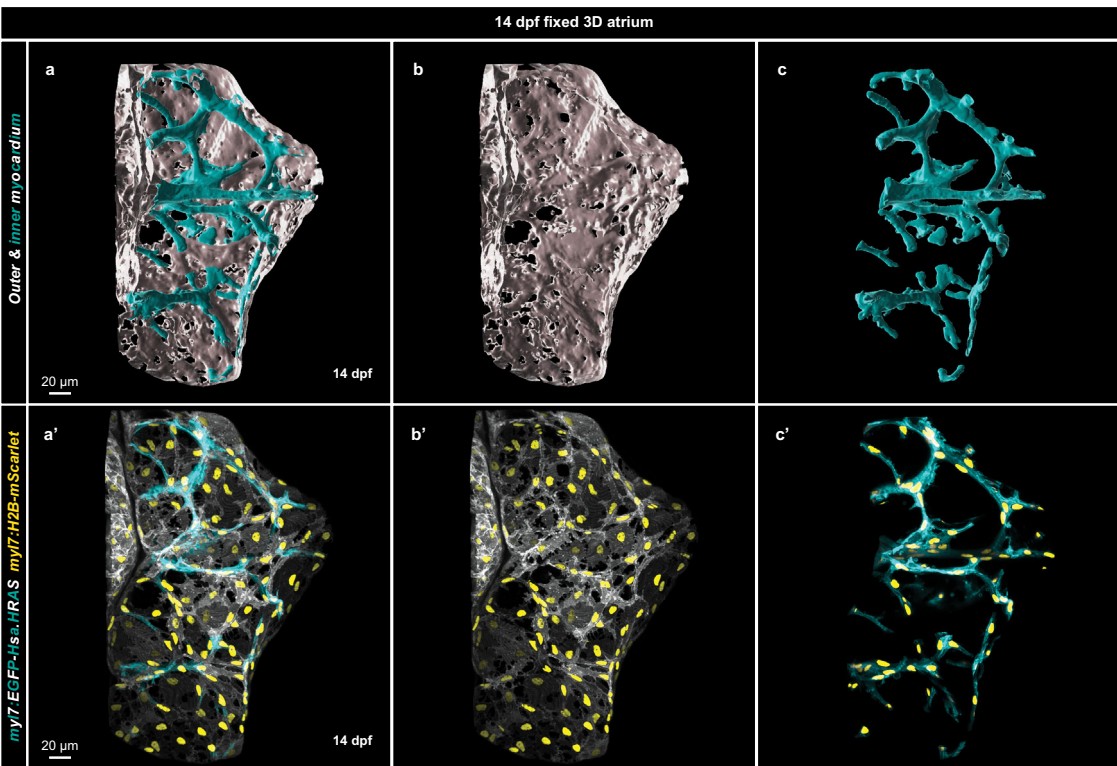

**Fig. 3 | Elongated cardiomyocytes form the inner atrial muscle structures.**
**a–c** 3D surface rendering of a 14 dpf fixed atrium; outer layer myocardium shown in white and inner layer myocardium in cyan. **a'–c'** 3D confocal image of the atrium from which the surface rendering was created; CM membranes shown in white for outer layer CMs and in cyan for inner layer CMs; all CM nuclei are shown in yellow.

that prevents premature elongation. Surprisingly, even though MYL9$^{CA}$ was present in CMs from an early stage, as assessed by EGFP expression, atrial CM elongation was only observed from 76 hpf onwards, in both control and MYL9$^{CA}$ overexpressing larvae (Supplementary Fig. 4e–k). However, the number of elongating atrial CMs in both 76 and 100 hpf MYL9$^{CA}$ overexpressing larvae was at least doubled compared with control (Supplementary Fig. 4e-k).

Next, we hypothesized that the formation of small membrane protrusions prior to CM elongation was required for the emergence of larger oriented protrusions. To test this hypothesis, we determined the average number of membrane protrusions per CM in 76 hpf atria of wild-type, CM specific MYL9$^{DN}$ overexpressing (Supplementary Fig. 5a–c, Supplementary Table 1), and IRSp53$^{DN}$ overexpressing (Supplementary Fig. 5d–f, Supplementary Table 1) larvae. No reduction in the number of small membrane protrusions was observed in MYL9$^{DN}$ overexpressing larvae (Supplementary Fig. 5a–c, Supplementary Table 1), suggesting that CM specific MYL9$^{DN}$ overexpression does not impair atrial CM elongation by inhibiting the formation of small protrusions. In contrast, IRSp53$^{DN}$ overexpression appeared to reduce the number of small membrane protrusions (Supplementary Fig. 5e, Table 1), perhaps leading to the observed impairment in both CM elongation and orientation (Fig. 4e–h"). Altogether, these data indicate that atrial CM elongation is an active process driven by membrane protrusion formation.

## Atrial morphogenesis is not regulated by factors important for ventricular trabeculation

Given the importance of the Notch and Nrg/ErbB signaling pathways during various aspects of cardiac morphogenesis including ventricular wall formation[51], we aimed to test whether they were also involved in atrial wall formation. We first searched for Notch signaling pathway activity in the atrium during atrial CM elongation, but observed no Notch reporter expression in the atrium between 100 and 148 hpf

(Supplementary Fig. 6a–c). Consistent with these data, the number of elongating atrial CMs and the process of atrial convergent thickening did not change significantly after Notch inhibitor treatment between 72 and 124 hpf (Supplementary Figs. 6d–f). Furthermore, we observed no significant differences in the number of elongating CMs or in the formation of inner myocardial ridges in the atrium of 124 hpf *erbb2* mutants compared with homozygous wild-type siblings (Supplementary Fig. 6g–i), despite the presence of *nrg2a* expression in the larval atrium[46]. We also tested the role of cardiac contractility since it is also required for various aspects of cardiac morphogenesis including ventricular wall formation[41,54], but the number of elongating CMs or the formation of inner myocardial ridges in the atrium were not significantly affected by decreased contractility following BDM treatment from 96 to 124 hpf (Supplementary Fig. 6j–l). Lack of atrial CM contractions in *amhc/myh6* mutants[72] also did not block atrial CM elongation or intercalation, but slightly reduced the percentage of elongating atrial CMs (Supplementary Fig. 6m–o). However, the abnormal expansion of the atrial chamber in BDM-treated larvae and in *myh6* mutants could itself lead to atrial CM elongation, and thus single cell resolution approaches will be required to further address the role of atrial CM contractility in their elongation. Altogether, these data indicate that the key regulators of ventricular trabeculation such as the Notch and Nrg/ErbB signaling pathways are not required for atrial morphogenesis.

## Atrial morphogenesis is regulated through Yap activity

To uncover the molecular cues underlying atrial CM elongation, which is clearly visible by 100 hpf (Fig. 1), we isolated 48 and 72 hpf atrial CMs through fluorescence activated cell sorting (FACS), and performed bulk RNA-sequencing (Supplementary Fig. 7). Unbiased pathway enrichment analysis of genes significantly upregulated at 72 hpf compared with 48 hpf uncovered genes promoting cell cycle progression (Supplementary Fig. 8a). We searched for the time point at which most

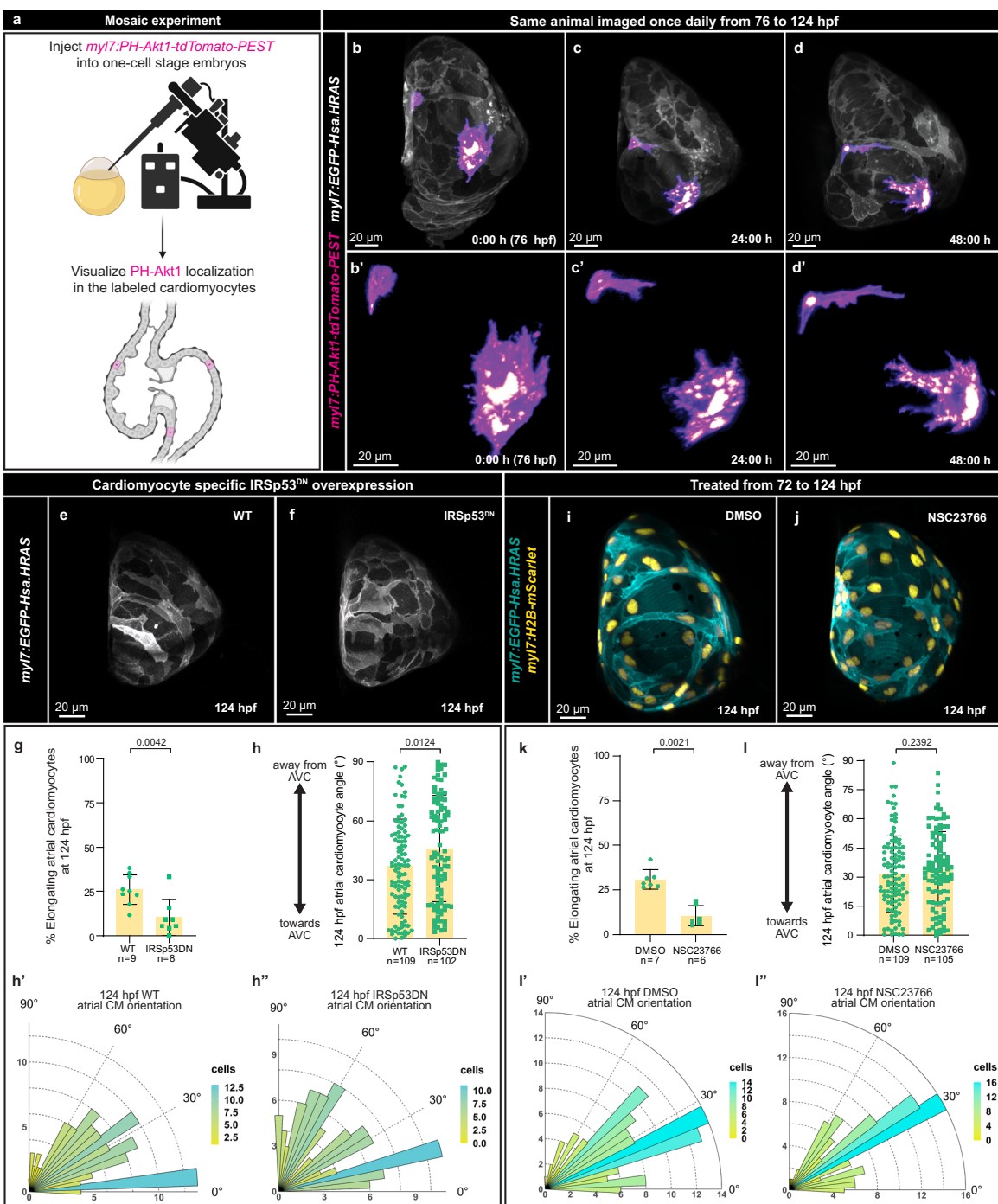

**Fig. 4 | Atrial cardiomyocytes elongate through membrane protrusion formation. a** Schematic of the membrane protrusion labeling of a few CMs, i.e., mosaic expression of the PH domain of Akt1 in CMs. Created with BioRender.com, released under a Creative Commons Attribution-NonCommercial-NoDerivs 4.0 International license. **b**–**d′** 3D airyscan imaging at 76, 100, and 124 hpf; CM membranes shown in white (*myl7:EGFP-Hsa.HRAS*) and PH-Akt1 localization in magenta (*myl7:PH-Akt1-tdTomato-PEST*). **e**, **f** 3D confocal images of the atrium from control sibling and IRSp53^DN overexpressing (*myl7:gal4; UAS:IRSp53DN-RFP*) larvae at 124 hpf; CM membranes shown in white (*myl7:EGFP-Hsa.HRAS*). **g** Percentage of elongating atrial CMs in 124 hpf control sibling and IRSp53^DN overexpressing larvae (*n* = 9 control and *n* = 8 IRSp53^DN; each data point represents one heart; two-tailed Mann-Whitney test). **h**–**h″**, Atrial CM angle measured in 124 hpf control siblings and

IRSp53^DN overexpressing larvae (*n* = 109 atrial CMs from 9 control hearts and *n* = 102 atrial CMs from 8 IRSp53^DN overexpressing hearts; each data point represents one atrial CM; two-tailed Mann-Whitney test). **i**, **j** 3D confocal images of the atrium from DMSO-treated (**i**) and Rac1 inhibitor-treated (**j**) larvae at 124 hpf; CM membranes shown in cyan (*myl7:EGFP-Hsa.HRAS*) and CM nuclei in yellow (*myl7:H2B-mScarlet*). **k** Percentage of elongating atrial CMs in DMSO- or Rac1 inhibitor-treated larvae at 124 hpf (*n* = 7 DMSO and *n* = 6 Rac1 inhibitor; each data point represents one heart; two-tailed Mann-Whitney test). **l**–**l″**, Atrial CM angle measured in 124 hpf DMSO-treated and Rac1 inhibitor-treated larvae (*n* = 109 atrial CMs from 7 DMSO hearts and *n* = 105 atrial CMs from 6 Rac1 inhibitor hearts; each data point represents one heart; two-tailed Mann-Whitney test). Error bars are mean ± SD.

atrial CMs enter S-phase by using a 24-hour EdU pulse assay starting at 48, 72, 96, and 120 hpf (Supplementary Fig. 8b). Interestingly, we found the highest proportion of EdU[+] atrial CMs at 96 hpf (Supplementary Fig. 8b). Furthermore, we observed a significant increase in the number of atrial CMs at 124 hpf (Supplementary Fig. 8c). Altogether, these data suggest that at 72 hpf, atrial CMs upregulate the expression of genes that promote cell cycle progression and subsequently proliferate. As such, the peaks of atrial CM elongation (Fig. 1f) and atrial CM proliferation (Supplementary Fig. 8c) coincide at 124 hpf.

The Hippo pathway is a well-known regulator of cell proliferation and tissue morphogenesis[73], and several of the proliferation genes significantly upregulated at 72 hpf, including *pcna*, *mcm* genes, *pola2*, *cdk1*, *ccna2*, *trip13*, *orc5*, *esco2*, and *cdkn1a* are known Hippo targets[74,75] (Supplementary Fig. 8a). During zebrafish cardiac development, Hippo pathway inhibition through *lats1* and *lats2* inactivation led to an increase in atrial CM numbers[76]. Thus, we hypothesized that the increase in atrial CM proliferation observed in wild-type larvae after 74 hpf occurs through changes in Hippo pathway activity. Consistent with this hypothesis, we observed expression of Hippo pathway genes including the transcriptional effector genes *yap1* and *wwtr1* in atrial CMs at both 48 and 72 hpf (Supplementary Fig. 8d). We then used a Yap1 antibody, previously validated during zebrafish cardiac development[77,78], and observed immunostaining in the nucleus of approximately 10% of atrial CMs at 76 hpf, which increased to approximately 25% at 100 hpf, before decreasing to approximately 15% at 124 hpf (Supplementary Figs. 8e–f′). However, nuclear Yap1 immunostaining was not clearly observed in elongating atrial CMs (6 hearts examined at 124 hpf) (Supplementary Fig. 8g, g′).

We then tested whether Yap1 was required for atrial morphogenesis. We incubated larvae from 72 to 124 hpf with K-975 or IWR-1, two inhibitors previously used in zebrafish[79–81]. K-975 blocks the interaction between Yap and its co-activator Tead[82], and IWR-1 is a tankyrase inhibitor that can sequester Yap inside the cytoplasm[83]. Both K-975 and IWR-1 treatments reduced atrial CM elongation (Fig. 5a–h″). Although IWR-1 can also inhibit the Wnt/β-catenin signaling pathway, we did not observe Wnt/β-catenin reporter activity in atrial CMs (Supplementary Fig. 9a–c), and overexpression of the Wnt/β-catenin pathway inhibitor Axin1 in atrial CMs did not affect atrial CM elongation significantly (Supplementary Fig. 9d–f).

At the cellular level, atrial CM elongation is an active process driven by cytoskeletal changes (Fig. 4). To investigate whether the Hippo pathway is required upstream or downstream of these cytoskeletal changes, we treated MYL9[CA] overexpressing larvae and control siblings with K-975 or IWR-1 and observed a rescue in the number of elongating atrial CMs in both treatment conditions (Supplementary Fig. 10a–g). However, K-975 or IWR-1 treatments did not appear to have an effect on the ability of atrial CMs to form small membrane protrusions (Supplementary Fig. 10h–k, Supplementary Table 1). Altogether, these data suggest that atrial CM elongation is regulated, at least in part, by the Hippo pathway, and that the cytoskeletal changes important for CM elongation, but not for small membrane protrusion formation, occur downstream of the Hippo pathway.

Hippo pathway inhibition negatively affected atrial CM elongation, therefore we tested whether this inhibition would also affect cardiac function (Supplementary Fig. 3). K-975 and IWR-1 treatments blocked the increase in myofibril thickness and alignment observed in control larvae (Supplementary Fig. 11a–f). Analysis of the inner atrial myocardial surface of DMSO- versus K-975- or IWR-1-treated larvae revealed the loss of the complex ridges at 124 hpf (Supplementary Fig. 11h–j). Furthermore, the atrial ejection fraction (Supplementary Fig. 11g), but not the heart rate (Supplementary Fig. 11k), was significantly reduced upon both treatments, suggesting that the decreased in atrial CM intercalation and in myofibril organization affects atrial chamber contraction.

## Yap1 establishes a cell heterogeneity important for atrial morphogenesis

To investigate how Yap1 regulates atrial CM elongation at the single cell level, we took advantage of a DN form of zebrafish Yap1 fused to a nuclear localization signal (NLS)[84,85]. We cloned this NLSYap1[DN] construct downstream of a heat shock-inducible promoter (*hsp70l:lox-TagBFP-STOP-lox-NLS-Yap1DN-t2a-mCherry*, abbreviated *hsp70l:LSL-NLSYap1DN*), and injected the resulting plasmid into *myh6:creERT2* embryos which allowed for atrial CM-specific expression upon cre-mediated recombination (Fig. 6a–c). Yap1[DN] overexpression, visualized by mCherry expression, was specifically present in atrial CMs in a mosaic fashion upon tamoxifen treatment and heat shock (Fig. 6c). Notably, quantification of the percentage of elongating CMs that were wild-type (TagBFP[+]) or Yap1[DN] positive (mCherry[+]) revealed more elongating CMs that overexpressed Yap1[DN] (Fig. 6d), suggesting that CMs with impaired Yap activity have a higher propensity to elongate. Thus, global inhibition of Yap activity reduced the number of elongating atrial CMs, while CMs with impaired Yap activity have a higher propensity to elongate. To investigate this apparent discrepancy, we looked more closely at atrial CM proliferation and observed that global inhibition of Yap activity also reduced atrial CM numbers (Supplementary Fig. 12a, b). To investigate the relationship between CM division and elongation, we incubated larvae from 72 to 124 hpf with the DNA replication inhibitor Aphidicolin, which resulted in a reduction in atrial CM numbers (Supplementary Fig. 12c), consistent with previous data[86,87]. However, this treatment did not affect the absolute number of elongating atrial CMs, thereby leading to an increase in their percentage (Supplementary Fig. 12d). These data lead us to suggest that after 76 hpf the atrial myocardium becomes heterogeneous with at least two types of atrial CMs, round Yap1[+] CMs and elongating Yap1[-] CMs, and that Yap1 regulates atrial CM elongation independently of its effect on cell division. To test this model, we generated a stable *hsp70l:LSL-NLSyap1DN* transgenic line using the same construct previously used for the mosaic experiment (Fig. 6a–d). Global overexpression of *DNyap1* at early developmental stages using this new line caused cardiac looping defects (Fig. 6e, f), consistent with previous studies of early loss of Yap1 function in zebrafish[85,88,89]. We then overexpressed DNYap1 in all atrial CMs, after recombination with the *myh6:creERT2* transgene, and observed a significant decrease in the number of elongating atrial CMs at 124 hpf (Fig. 6g–i), as previously observed when inhibiting Yap1 function pharmacologically (Fig. 5). Altogether, these data indicate that Yap1 is involved in atrial CM elongation.

## Distinct mechanisms drive atrial and ventricular muscle structure formation

Since Hippo pathway activity appears to be required for atrial morphogenesis, and Taz/Wwtr1 is important during ventricular wall formation[90], we hypothesized that the Hippo pathway was regulating both atrial and ventricular muscle structure formation. To test this hypothesis, we imaged the atrium of *wwtr1* mutants and homozygous wild-type siblings at 124 hpf and observed no atrial morphogenesis defects (Fig. 7a–c). Conversely, K-975, IWR-1, or Rac1 inhibitor treatment did not block the onset of trabeculation (Fig. 7d–h). Together these data suggest that the onset of atrial and ventricular muscle structure formation occurs through distinct cellular and molecular mechanisms (Fig. 7i).

## Discussion

Our high-resolution 3D live imaging revealed that starting at early larval stages, a subset of atrial CMs extend membrane protrusions leading to their elongated appearance. This CM elongation leads to cell intercalation and convergent thickening of the atrial myocardium, ultimately resulting in the formation of the inner muscle network.

Atrial CM elongation could be an indirect effect of chamber ballooning[36,91] or it could represent an active process. Our observations

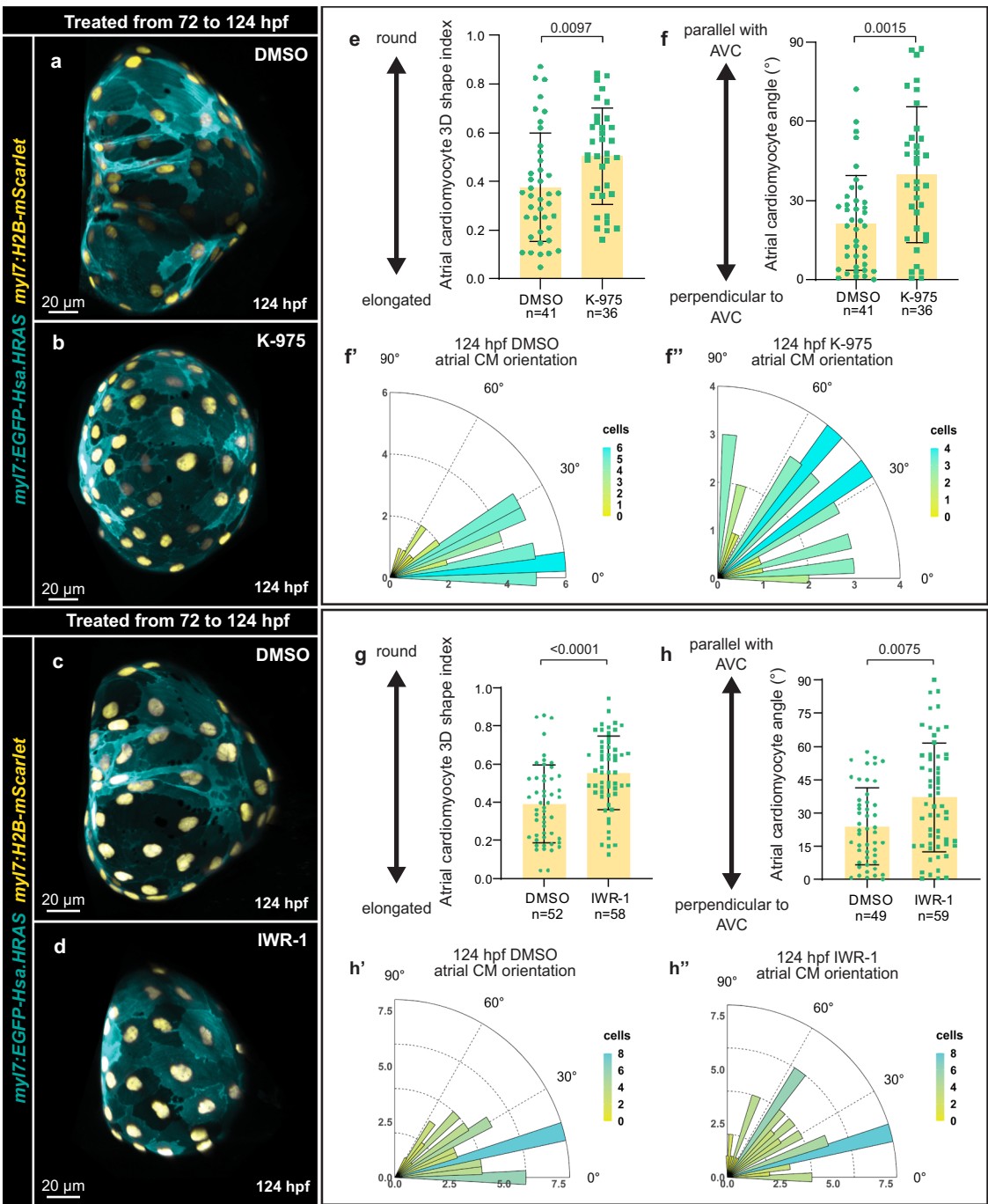

**Fig. 5 | Hippo pathway inhibition affects atrial cardiomyocyte elongation.**
**a**–**d** 3D airyscan images of atria from 124 hpf larvae treated from 72 to 124 hpf with DMSO (**a**, **c**), K-975 (**b**), or IWR-1(**d**); CM membranes shown in cyan (*myl7:EGFP-Hsa.HRAS*) and CM nuclei in yellow (*myl7:H2B-mScarlet*). **e**, **g** 3D quantification of atrial CM shape in 124 hpf larvae treated from 72 to 124 hpf with DMSO, K-975, or IWR-1 (*n* = 41 atrial CMs from 4 DMSO hearts, *n* = 36 atrial CMs from K-975 hearts, *n* = 52 atrial CMs from 5 DMSO hearts, and *n* = 58 atrial CMs from 5 IWR-1 hearts;

each data point represents one atrial CM; average of 10 CMs per heart two-tailed Mann-Whitney test (**e**); two-tailed unpaired Student's t-test *p*-value = 0.48*10⁻⁴ (**g**). **f**–**f″**, **h**–**h″**, Atrial CM angle measured in 124 hpf larvae treated from 72 to 124 hpf with DMSO, K-975, or IWR-1 (*n* = n = 41 atrial CMs from 4 DMSO hearts, *n* = 36 atrial CMs from 4 K-975 hearts, *n* = 49 atrial CMs from 5 DMSO hearts, and *n* = 59 atrial CMs from 5 IWR-1 hearts; each data point represents one atrial CM; average of 10 CMs per heart; two-tailed Mann-Whitney tests). Error bars are mean ± SD.

and manipulations of atrial CM protrusive activity indicate that it is an active process. These atrial CMs elongate along the long axis of the heart and in both directions, i.e., towards the AVC and the sinoatrial node. Thus, these specialized cardiac populations[92,93] could provide signals to stimulate atrial CM protrusive activity and/or directionality. Along these lines, we tested the role of Wnt/β-catenin activity, which is present in the AVC and sinoatrial node[92,93], in atrial CM elongation but failed to see an effect. Interestingly, only a subset of atrial CMs

elongate raising important questions as to how they are selected and what limits their number. Furthermore, not all elongating atrial CMs end up in the inner muscle structures as some remain in the outer layer at least up to 14 dpf. The cellular and molecular mechanisms that select the CMs that form the inner ridges versus those that form the outer layer, as well as the physiological relevance of having an outer layer composed of mixed CM morphology, remain to be investigated. It is possible that maintaining the round atrial CM population is required

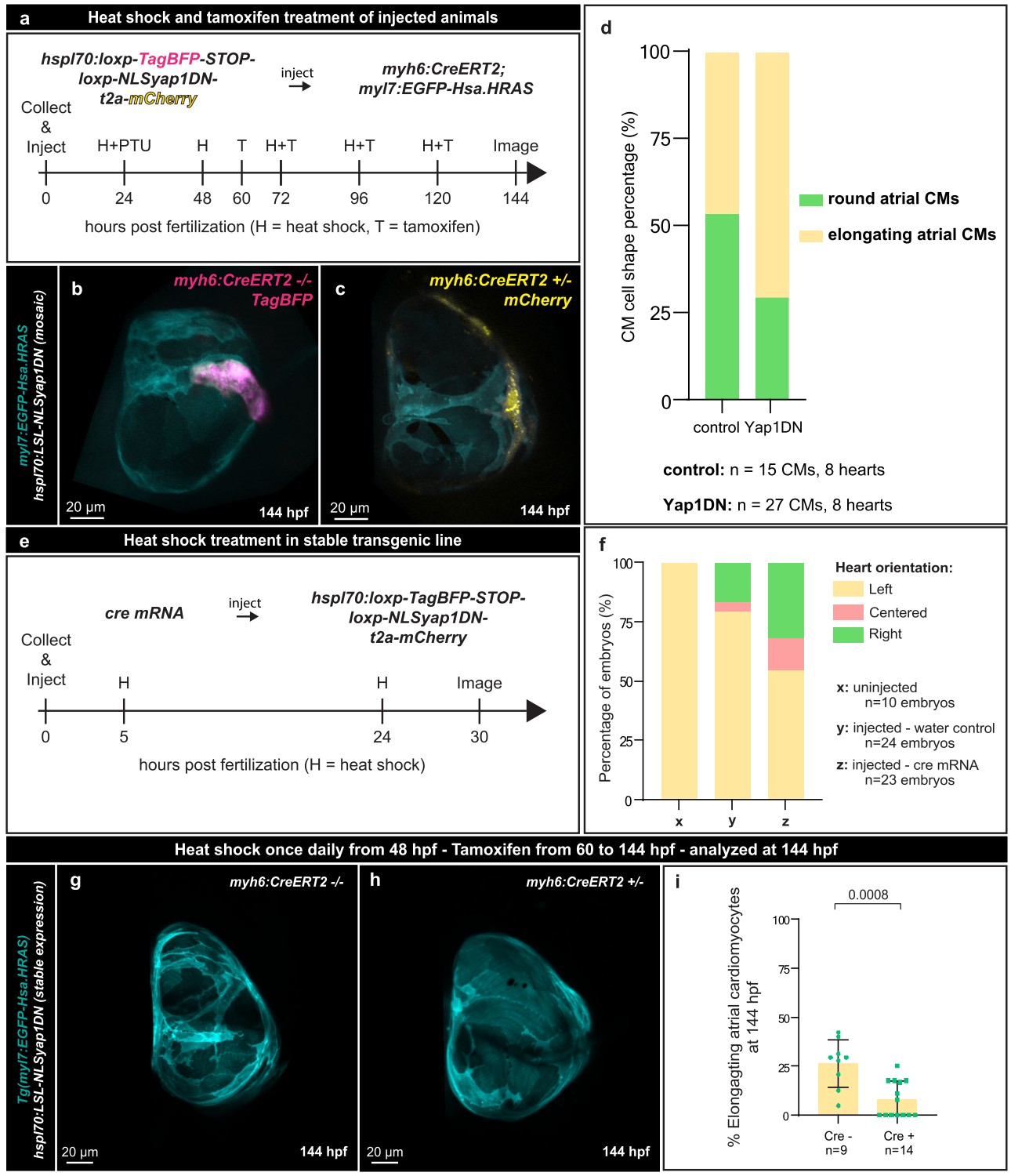

**Fig. 6 | Yap modulates atrial cardiomyocyte elongation. a** Schematic of mosaic Yap1DN overexpression in CMs. **b**, **c** 3D confocal images of 144 hpf atria from control mosaic labeled (**b**) (creERT2⁻, TagBFP⁺) and Yap1DN mosaic overexpressing (**c**) (creERT2⁺, mCherry⁺) larvae; CM membranes shown in cyan (*myl7:EGFP-Hsa.HRAS*), control mosaic labeling in magenta (*hspl70:LSL-NLSyap1DN, myh6:creERT2⁻*), and Yap1DN mosaic labeling in yellow (*hspl70:LSL-NLSyap1DN, myh6:creERT2⁺*). **d** Percentage of elongating CMs from control labeled or Yap1DN overexpressing CMs (*n* = 8 hearts, 15 control labeled CMs; *n* = 8 hearts, 27 YAP1DN overexpressing CMs). **e** schematic of Yap1DN overexpression using the stable transgenic line. **f** Percentage of left, centered, and right oriented hearts at 30 hpf from uninjected,

water injected, and cre mRNA injected *hspl70:LSL-NLSyap1DN* embryos (*n* = 10 uninjected controls, *n* = 24 water injected controls, and n = 23 *cre* mRNA injected embryos). **g**, **h** 3D airyscan images of 144 hpf atria from *hspl70:LSL-NLSyap1DN* control larvae (**g**) and *hspl70:LSL-NLSyap1DN; myh6:CreERT2* siblings (**h**); CM membranes shown in cyan (*myl7:EGFP-Hsa.HRAS*); larvae were heat shocked once daily from 48 hpf and kept in Tamoxifen from 60 to 144 hpf, refreshed once daily. **i**, Percentage of elongating atrial CMs at 144 from *hspl70:LSL-NLSyap1DN* control larvae and *hspl70:LSL-NLSyap1DN; myh6:CreERT2* siblings (*n* = 9 Cre⁻ controls and *n* = 14 Cre⁺ siblings, two-tailed Mann-Whitney test). Error bars are mean ± SD.

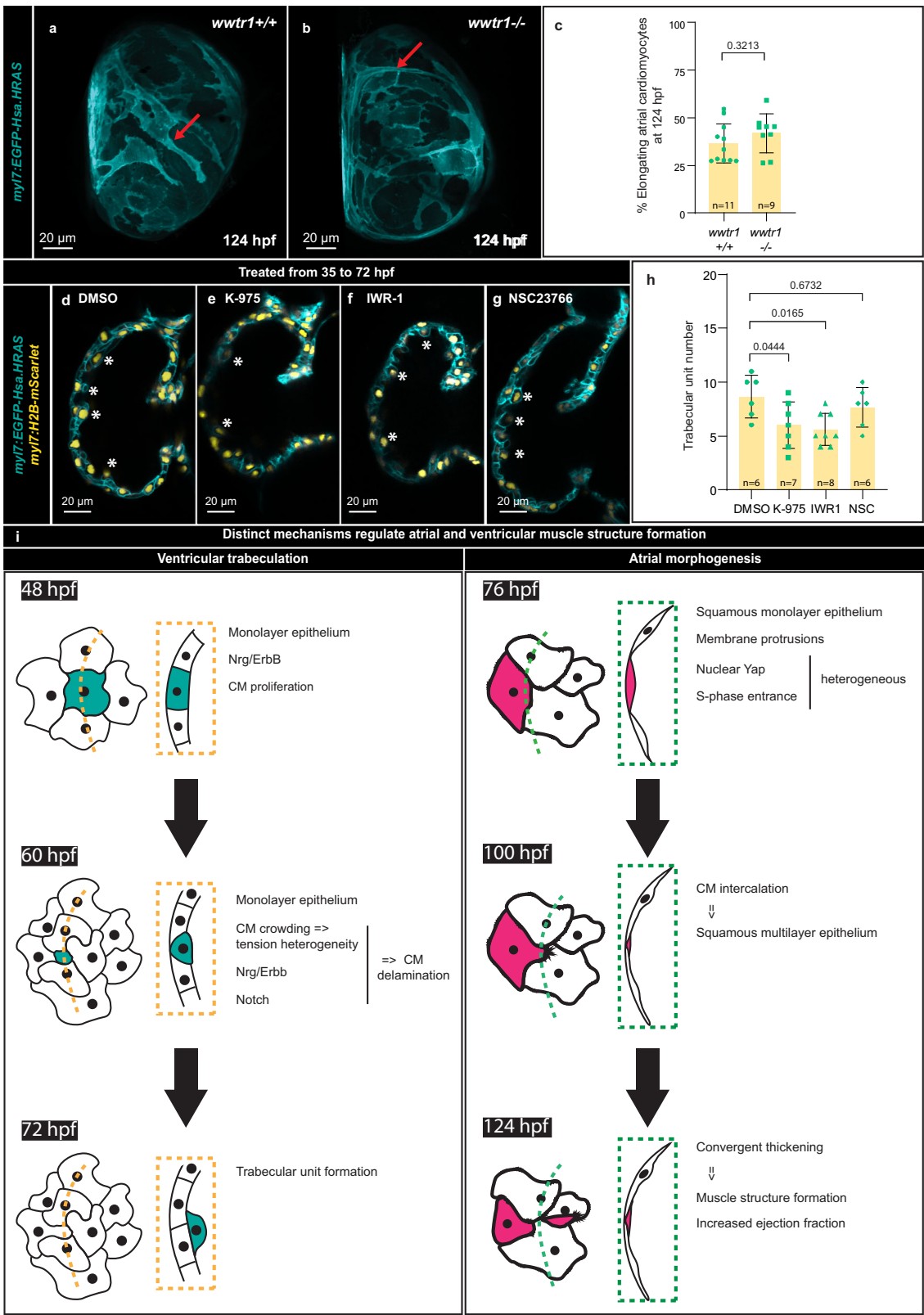

**Fig. 7 | Distinct mechanisms regulate atrial and ventricular muscle structure formation. a, b** 3D airyscan images of atria from homozygous wild-type siblings and *wwtr1* mutants at 124 hpf; CM membranes shown in cyan (*myl7:EGFP-Hsa.H-RAS*). **c** Percentage of elongating atrial CMs in 124 hpf homozygous wild-type siblings and *wwtr1* mutants (*n* = 11 wild types, *n* = 9 mutants; each data point represents one heart; two-tailed Mann-Whitney test). Error bars are mean ± SD. **d–g** 2D confocal planes of ventricles from 72 hpf larvae treated from 35 to 72 hpf with DMSO, K-975, IWR-1, or Rac1 inhibitor; CM membranes shown in cyan

(*myl7:EGFP-Hsa.HRAS*) and nuclei in yellow (*myl7:H2B-mScarlet*). **h** Number of trabecular units in 72 hpf larvae treated from 35 to 72 hpf with DMSO, K-975, IWR-1, or Rac1 inhibitor (*n* = 5 DMSO, *n* = 7 K-975, *n* = 8 IWR-1, *n* = 6 Rac1 inhibitor; each data point represents one heart; ordinary one-way ANOVA with Dunnett's multiple comparison test); red arrows point to elongating CMs; white asterisks indicate trabecular units. Error bars are mean ± SD. **i** Schematic summarizing the cellular and molecular processes involved in ventricular (left) and atrial (right) wall morphogenesis (manually rendered using Adobe Illustrator).

for the integrity of the tissue as neighboring CMs elongate and intercalate.

While Notch signaling is important in limiting the number of ventricular CMs that delaminate[49], it doesn't appear to play a similar role in atrial CM selection (Supplementary Fig. 6a–f). Conversely, global inhibition of Yap activity reduced atrial CM elongation but it did not stop the onset of ventricular trabeculation. However, inhibiting Yap activity resulted in a reduced number of trabecular units, and *wwtr1* mutants also exhibit trabeculation defects[90], suggesting that the Hippo signaling pathway is involved in ventricular wall formation, either by modulating CM behavior directly and/or indirectly via the endocardium or epicardium.

Concurrent with atrial CM elongation, we observed atrial CM proliferation; and while these processes appear to be mutually exclusive (as suggested by the data from the Aphidicolin treatments), pharmacological inhibition of Yap activity led to a decrease of both processes. In addition, we observed Yap1 in the nuclei of a subset of atrial CMs, and mosaic Yap1[DN] overexpression in atrial CMs cell-autonomously promoted their elongation, whereas overexpressing Yap1[DN] in all atrial CMs using a stable line impaired their elongation. Possible scenarios to explain all these observations include the need for heterogenous Yap1 activation within the atrial myocardium to allow for atrial CM elongation. Yap has been implicated in several different developmental processes including cell proliferation and cell migration[94], and more sophisticated tools and approaches will be required to dissect its roles in atrial morphogenesis.

Following CM elongation, cell intercalation and convergent thickening of the atrial myocardium leads to the transformation of a monolayered epithelium into a complex structure. Convergent thickening was originally described to occur within the mesoderm prior to involution during amphibian gastrulation[95]. While the cellular and molecular basis of convergent thickening remains to be discovered, it was recently reported to be independent of dorsoventral patterning[61]. Notably, similar to atrial CM intercalation, convergent thickening during gastrulation involves the formation of oriented lamellipodia under the control of cell contractility[96], and it will be interesting to investigate how similar convergent thickening is in these two different developmental settings, as well as during other morphological processes.

Ultimately, the ventricular and atrial chambers play distinct roles; while the ventricles need to build strong muscular walls to propel the blood throughout the body, the atria need to provide a structure that can collect blood and propagate the action potential quickly. The elongation of some atrial CMs may provide the means to achieve this quick impulse propagation.

## Methods

### Zebrafish handling

Zebrafish larvae were raised under standard conditions. All transgenic larvae were presorted for fluorescence. Non-mutant larvae were screened for the presence of developmental defects (e.g., pericardial edema, situs inversus, cardiac size or shape abnormalities) prior to experiments and excluded from the pool if they displayed any.

Adult fish were maintained in 3.5 l tanks at a stock density of 10 fish/l with the following parameters: water temperature: 27–27.5 °C; light:dark cycle: 14:10; pH: 7.0–7.5; conductivity: 750-800 μS/cm. Fish were fed 3–5 times a day, depending on age, with granular and live food (*Artemia salina*). Health monitoring was performed at least once a year. All procedures performed on animals conform to the guidelines from Directive 2010/63/EU of the European Parliament on the protection of animals used for scientific purposes and were approved by the Animal Protection Committee (Tierschutzkommission) of the Regierungspräsidium Darmstadt (reference: B2/2057). Sample size was calculated to minimize the number of animals used in compliance to the principle of the 3Rs and the European Directive 2010/63/EU.

### Zebrafish lines

The transgenic and mutant lines used in this study are: *Tg(myl7:EGFP-Hsa.HRAS)s883*[97], abbreviated *myl7:EGFP-Hsa.HRAS*; *Tg(-0.8myl7:H2B-mScarlet)bns534*[98], abbreviated *myl7:H2B-mScarlet*; *TgBAC(cdh2:cdh2-EGFP,crybb1:ECFP)zf517*[99], abbreviated *cdh2:cdh2-EGFP*; *Tg(myl7:m-Cherry-CAAX)bns7*[48], abbreviated *myl7:mCherry-CAAX*; *Tg(UAS: AS3DN-p2a-tagRFP)bns440*[67], abbreviated *IRSp53[DN]*; *Tg2(myl7:GAL4)cbg2Tg*[68], abbreviated *myl7:gal4*; *Tg(myl7:actn3b-EGFP)sd10*[100], referred to as *myl7:actinin3b-EGFP*; *Tg(myl7:lck-mScarlet)bns561* (this study), abbreviated *myl7:lck-mScarlet*; *Tg(myl7:Hsa.MYL9_T19D,S20D-EGFP)bns332*[49], abbreviated *myl7: MYL9CA-EGFP*; *Tg(myl7:Hsa.MYL9_T19A,S20A-EGFP) bns333*[49], abbreviated *myl7: MYL9DN-EGFP*; *Tg(myl7:LIFEACT-GFP) s974*[40] abbreviated *myl7:LIFEACT-GFP*; *Tg(7XTCF-Xla.Siam:GFP)ia4*[101], abbreviated *7XTCF-Siam:GFP*; *Tg(myh7:mCherry-Eco.NfsB)s957*[102], abbreviated *myh7:mCherry-NTR*; *Tg(-4.5myh6:mTagBFP2)bns629* (this study), abbreviated *myh6:mTagBFP2*; *Tg(hsp70l:LOXP-Luciferase-MYC-STOP-LOXP-Mmu.-Axin1-NLS-dTomato,cryaa:AmCyan)ulm14*[103], abbreviated *hsp70l:LSL-Mmu.Axin1-p2a-NLS-dTomato*; *Tg(myl7:Cre)sd55*[43], abbreviated *myl7:Cre*; *Tg(myh6:CreERT2)sd20*[102], abbreviated *myh6:-CreERT2*; *Tg(EPV.Tp1-Mmu.Hbb:Venus-Mmu.Odc1)s940*[104], abbreviated *Tp1-Hbb:Venus-Mmu.Odc1*; *Tg(hsp70l:loxP-TagBFP-STOP-loxP-NLS-DNyap1-T2A-mCherry)bns729* (this study), abbreviated *hsp70l:LSL-NLSyap1DN*; *erbb2[st61]* (ref[105].); *wwtr1[bns35]* (ref[106].); *myh6[s812]* (ref[72].).

### Plasmid construction

For the membrane localizing *myl7:lck-mScarlet* construct, the insert was generated through primer annealing (forward primer 5′-GCGGCCATCGAATTGGGATCCCCACCATGGGCTGCGTGTGCAGCAG-CAACCCCGAG-3′; reverse primer 5′-GCCCTTGCTCACGGGTACCACCG GTCGCTCGGGGTTGCTGCTGCACACGCAGCCCAT-3′). The Tol2 plasmid *-0.8myl7:H2B-mScarlet*[98] was prepared through BamHI/AgeI restriction digest and the *lck* insert was inserted. To generate the atrial CM cytoplasmic construct *-4.5myh6:mTagBFP2*, the *mTagBFP2* insert was PCR amplified (forward primer 5′-TCTAGAGCGGCCATCGAAT TGGGGATCCCCACCATGGTGTCTAAG-3′; reverse primer 5′-GCCGC CAGTGTGATGGATATCTTAATTAAGCTTGTGCCCCAGTTGCTAG-3′) from *-0.8myl7:mTagBFP2* (gifted from Dr. Rashmi Priya) and inserted into a Tol2 vector containing the *-4.5myh6* promoter (gifted from Dr. Thomas Juan). To create the Yap1[DN] inducible construct *hsp70l:loxP-TagBFP-STOP-loxP-NLS-yap1DN-t2A-mCherry*, a Tol2 plasmid *hsp70l:loxP-TagBFP-STOP-loxP-cxcl18b-t2A-mCherry* (gifted from Pinelopi Goumenaki), was digested with XhoI/AgeI to remove the *cxcl18b* cassette. The pCS2 *NLSyap1DN* construct[85] (gifted from Prof. Virginie Lecaudey) was PCR amplified (forward primer 5′-ATTATACGAAGTTA-TACCGGTATGGCTCCAAAGAAGAAGCG-3′; reverse primer 5′-GCCCTCTCCACTGCCCTCGAGCCTCAGTCTCTCCTTCTCTA-3′), and inserted in the inducible vector. All fusion experiments were performed using a 5X In-Fusion HD Enzyme Premix (Takara, ST0345).

### Zebrafish transgenesis

The *-0.8myl7:mTagBFP2* (CM cytoplasmic marker), *myl7:PH-Akt1-tdTomato-PEST* (CM membrane protrusion marker), and *hsp70l:loxP-TagBFP-STOP-loxP-NLSyap1DN-t2A-mCherry* (inducible Yap1[DN] over-expression) Tol2 plasmids were injected together with *Tol2* mRNA into one-cell stage wild-type zebrafish embryos for imaging of mosaic hearts (Fig. 4a). The *-0.8myl7:lck-mScarlet*, *-4.5myh6:mTagBFP2*, and *hsp70l:loxP-TagBFP-STOP-loxP-NLS-yap1DN-t2A-mCherry* plasmids were injected together with *Tol2* mRNA into one-cell stage wild-type zebrafish embryos which were grown into adulthood. The positive offspring were outcrossed for 2 generations before establishing the stable line.

### Immunostaining

Larvae were immunostained as previously described[107]; however deyolking was skipped to maintain atrial integrity. In brief, larvae were

incubated in 0.1% Tricaine/egg water prior to fixation at 4 °C overnight in 4% paraformaldehyde, and then gently washed 3 times with PBS/0.1% Tween; after treatment for 1 hour in PBS/0.1% Tween/5 µg Proteinase K, the larvae were washed 3 times in PBS/1% BSA/1% DMSO/0.5% Triton-X (PBDT), and then blocked for at least 1 hour in PBDT/10% goat serum; primary antibodies were mixed with fresh blocking buffer and incubated with the larvae overnight at 4 °C. The Yap polyclonal antibody was preincubated at 4 °C with zebrafish larvae to improve the signal-noise ratio. Primary antibodies used: anti-Yap1 (rabbit polyclonal, a generous gift from Prof. Virginie Lecaudey, 1:200), anti-eGFP (chicken monoclonal, ab13970 Abcam, 1:200). Secondary antibodies used: Alexa Fluor 488, and Alexa Fluor 647 (1:500, Thermo Fisher Scientific, produced in goat).

### EdU staining

*Tg(myl7:H2B-mScarlet)* larvae were incubated for 24 h in 1-phenyl-2-thiourea (PTU) egg water supplemented with 500 µM EdU and 0.5% DMSO, starting from 48, 72, 96, or 120 hpf. After EdU incubation, the larvae were bathed in 0.2% Tricaine to stop the heart before fixing them in 4% PFA for 2 h at room temperature. Next, the 30 min click-it reaction was performed as per manufacturer's instructions (Invitrogen, C10340), and confocal imaging was carried out immediately after the washes.

### Larval treatments

Zebrafish larvae were incubated in 6-well plates, 6–7 larvae per well, in 4 ml of PTU egg water in which the respective stock reagent was diluted. The incubation time depended on the specific experiment. The solutions were refreshed every 24 h and the larvae were kept in the solution until imaging. The concentrations used in this study were selected based on concentration gradient tests, whereby we used the lowest possible concentration that induced a cardiac phenotype, which was always lower than the concentration that triggered overall toxicity. The control treatments used PTU egg water with the equivalent concentration of DMSO.

The pharmacological reagents used in this study are: NSC23766 (Rac1 inhibitor, incubated from 72 to 124 hpf at 150 µM, or from 35 to 72 hpf at 200 µM, Selleck chemicals S8031), K-975 (Yap/Tead inhibitor, incubated from 72 to 124 hpf at 4 µM, or from 72 to 100 hpf at 4 µM, or from 35 to 72 hpf at 3 µM, Selleck chemicals, E1329), IWR-1 (tankyrase inhibitor, incubated from 72 to 124 hpf at 10 µM, or from 72 to 100 hpf at 10 µM, or from 35 to 72 hpf at 12.5 µM, Selleck chemicals S7086), LY411575 (γ-secretase inhibitor, incubated from 72 to 124 hpf at 2.5 µM, Selleck chemicals S2714), 2,3-butanedione monoxime (BDM) (non-selective myosin ATPase inhibitor, incubated from 96 to 124 hpf at 7.5 mM, Sigma-Aldrich B0753), Aphidicolin (DNA polymerase α inhibitor, incubated from 72 to 124 hpf at 150 µM, Sigma-Aldrich A0781).

To activate the *hsp70l* promoter, larvae were incubated at 39 °C for 1 h, once daily. To induce CreERT2 nuclear translocation, larvae were treated with 10 µM 4-hydroxytamoxifen (4-OHT) (Sigma H7904, stock diluted in Ethanol) refreshed every 24 h.

### Imaging

To avoid pigmentation, for all imaging, embryos/larvae were treated with PTU starting at 24 hpf. To obtain 3D Airyscan images of the entire heart, larvae were mounted in 1% low-melting agarose and 0.2% Tricaine to both anesthetize and stop cardiac contractions during imaging. For longitudinal studies, the larvae were imaged for a maximum of 10 min after which they were carefully recovered in PTU egg water and placed in the incubator to restore cardiac contractions and development. The same larvae were imaged multiple times depending on the experiment (Fig. 1a). For the longitudinal tracking of 124 hpf to 14 dpf hearts, the larvae were imaged live at 124 hpf. The animals were then removed from the agarose and grown in standard conditions up to 14 dpf. At 14 dpf, the same larvae that were imaged at 124 hpf were

fixed in 4% PFA and their hearts dissected, mounted in 1% low-melting agarose, and imaged.

3D live images of the entire heart were obtained using the fast Airyscan mode from a Zeiss LSM880 Axio Examiner W Plan-Apochromat 20×/1.0 lens with the Zeiss ZEN software (ZEN 2.3 SP1 FP3 black). Images of fixed or live tissues where only the front of the atrium was required were taken using the confocal mode on a Zeiss LSM880 Axio Examiner W Plan-Apochromat 20×/1.0 lens with the Zeiss Zen software (ZEN 2.3 SP1 FP3 black), or a Zeiss LSM700 Axio Imager W Plan-Apochromat 40×/1.0 dipping lens with the Zeiss ZEN software (ZEN 2011 SP3 black).

Live imaging of the beating heart was obtained by mounting the larvae in 2% agarose without Tricaine to avoid potential arrythmogenic effects. To avoid cardiac contraction fluctuations caused by temperature changes, the embedded larvae were incubated in standard growth conditions for an additional 20 min before imaging using a preheated (27.5 °C) microscope incubator. Cardiac contractions were recorded with a Zeiss Spinning Disk confocal microscope with a 40x/1.1 W lens (ZEN 2.6 blue software), at 5 ms exposure for 20–30 sec. Light intensity and duration were kept at a minimum.

### Image processing

All images in the figures are representative images. All airyscan images were processed using ZEN (2.3 SP1 FP3 black), 3D airyscan processing mode (strength auto, 6.0). All 3D renderings, croppings and quantifications were performed using Imaris. The settings used to image the atrium caused a signal saturation in the ventricle (Fig. 1b); therefore, for better visualization, we used the 3D clipping tool in Imaris to exclude the ventricle from all subsequent images, except when the ventricle needed to be analyzed. 3D segmentation of the entire chamber and of single atrial CMs was rendered manually using the myocardial membrane marker as reference and the surface tool to draw, either the entire atrial chamber every 10 planes, or the cell boundaries every plane respectively. Total atrial CM numbers were determined by using the manually rendered atrial chamber surface as a mask to extract the atrial nuclear signal. The spot function was used to segment all atrial nuclei from the extracted signal and to obtain the total number of atrial CMs. The 3D shape index was measured using the 3D surfaces of single atrial CMs and the Ellipsoid-oblate equation available from Imaris (Supplementary Movie 1). To segment the inner atrial muscle structures in 3D, the inner membrane and nuclear signal was manually drawn for every plane to create a 3D mask which was used to extract the outer versus inner layer signal from the original one (Supplementary Movie 2). Both movies were rendered in this way using Imaris.

To obtain the atrial CM angle, the 3D images were first aligned in Imaris. For the alignment, the hearts were positioned such that the AVC CMs, which at these stages have an orientation perpendicular to that of the blood flow in the AV canal, were in a vertical position and the atrium and ventricle were kept parallel to each other (Fig. 1b). 2D images captured from this alignment were obtained using the Imaris Snapshot tool. These 2D images were further analyzed in ImageJ. Only the relatively flat CMs were analyzed and any CM on the curvature of the chamber was left out to avoid artefacts. The longest straight line was drawn through each analyzed CM and the angle of this line in reference to the AVC CM orientation was obtained as a measure of cell angle (Fig. 1b'). To obtain the average number of elongating CMs, the 2D images were first obtained through Imaris pre-alignment and the CMs located away from the curvature were scored for cell shape. To determine the number of small membrane protrusions formed by individual atrial CMs, 3D images were aligned in Imaris, and ImageJ was used to manually count the number of membrane protrusions for each CM. N-cadherin and membrane overlap longitudinal quantification was determined on pre-aligned 2D images. A line was drawn at the same position for each time point and the intensity plot was obtained.

The intensity plot revealed regions of high intensity at the membrane junctions and the diameter of these regions was obtained as a measure of the size of N-cadherin or membrane overlap. These diameters were normalized to the first measured time point and then plotted over time.

Myofibril quantifications were performed in ImageJ, on 2D images pre-aligned in Imaris, as described above, and only on myofibrils that were not positioned at the curvature of the chamber. Myofibril angle was measured by drawing a line parallel with the myofibril and the angle of this line in reference to the AVC CM angle was obtained as a measure of myofibril angle. Myofibril thickness was obtained by drawing a line perpendicular to myofibril orientation using the signal from the *myl7:actn3b-EGFP* Z-band reporter, perpendicular to myofibril orientation. For each wild-type heart, approximately 40 sarcomeres from at least 3 different cells of either round or elongated atrial CMs were averaged to obtain the average sarcomere thickness of the respective cell type. For hearts lacking atrial CM elongation, approximately 60 sarcomeres were measured from at least 6 different cells and averaged to obtain the average sarcomere thickness of that atrium. The percentage of CMs exhibiting nuclear Yap immunostaining was obtained by counting the number of atrial CMs with Yap signal colocalizing with the DAPI signal (nuclear marker) and with the CM membrane marker. This number was then compared to the total DAPI and GFP positive atrial CM nuclei. Heart rate kymographs and ejection fraction were also determined using ImageJ[26]. Data analysis blinding was only possible for mutant versus wild-type data, as the same person who imaged the other experiments (i.e., the wild-type and the pharmacological ones) also performed them.

## Atrial cardiomyocyte isolation and Fluorescence Activated Cell Sorting

To isolate atrial CMs, we used reporter lines that label atrial (*myh6:mTagBFP2*) and ventricular (*myh7:mCherry-NTR*) CMs, and a Wnt/β-catenin reporter (*7XTCF-Sia:eGFP*) that labels the AVC[108] and sinus venosus[109] CMs (Supplementary Figs. 5a–d'). We were thus able to reliably isolate atrial CMs by sorting the cells positive for mTagBFP2 and negative for eGFP and mCherry (Supplementary Fig 5e-h").

Approximately 150 hearts at 48 and 72 hpf were extracted through manual dissection and dissociated using the Pierce Cardiomyocyte Isolation Kit (Thermo Fisher Scientific, Catalog# 88281) as previously described[98]. In brief, hearts were dissected in Dulbecco's modified Eagle's medium (DMEM) + GlutaMAX (Thermo Fisher Scientific, Catalog# 10566016) supplemented with 10% FBS and kept on ice throughout the dissociation protocol; the hearts were centrifuged at 4 °C for 5 min at 2300 g; the supernatant was removed and the hearts were washed with 1 ml Hank's Balanced Salt Solution (HBSS); the tissue was dissociated into single cells by incubating it with 100 μl Enzyme 1 and 5 μl Enzyme 2 from the Pierce Cardiomyocyte Isolation Kit at 30 °C on a shaker set at 300 rpm; 1 ml of DMEM with FBS was added to stop the digestion and removed after centrifugation at 4 °C for 3 min at 800 g; fresh DMEM with FBS was added to resuspend the cells and pass them through a 40 μl-filtered fluorescence-activated cell sorting (FACS) sample tube. 2 μl of DRAQ7™ Dye (Catalog# D15106) was added and incubated in the dark for 10 min at RT. The cell suspension was filtered through a 35 μm nylon Falcon® 5 mL Round Bottom Polystyrene 12 × 75 mm Test Tube (Product# 352235). Cells were sorted using a BD FAC-SAria™ III (BD Biosciences) or an Invitrogen Bigfoot Spectral Cell Sorter (ThermoFisher Scientific) equipped with a 100 μM nozzle and with 20 psi pressure on the instruments. Live and non-AVC and non–SAN CMs were gated by exclusion of DRAQ7™ Dye using 633 nm excitation paired with 730/45 nm band pass filter and eGFP fluorescence using 488 nm excitation paired with 530/30 nm band

pass filter, respectively. To sort atrial CMs (*myh6*:mTagBFP2⁺), tagBFP fluorescence was measured with 405 nm excitation paired with 455/14 nm band pass filter or 450/50 nm band pass filter; to sort ventricular CMs (*myh7*:mCherry-NTR⁺), mCherry fluorescence was measured with 561 nm excitation paired with 610/20 nm band pass filter. Sorted cells were resuspended in 500 μl Trizol for subsequent RNA extraction. Cytometric data were recorded using FACSDiva software (Version 8.0.1; BD Biosciences), and Sasquatch Software (Version 1.19.2; ThermoFisher Scientific). Data analysis was performed using FlowJo software (Version 10.8, BD Biosciences).

Embryonic zebrafish atrial CMs are low in number and sensitive to dissociation, so in order to get enough cells for RNA-sequencing, a total of 5 sorting sessions were completed to obtain 1000-2000 cells per replicate, 3 replicates per condition. The first two replicates consist of cells sorted using the BD FACSAria™ III and BigFoot and the third replicate consists of cells sorted using the BigFoot only.

For gating (Supplementary Fig. 5e–h"), following exclusion of debris, the cell population was selected (Cells), from which single cells were gated using a FSC-A vs FSC-H parameter (Single cells). Within this single cell population, live cells were selected by exclusion of the DRAQ7™ Dye⁺ population and eGFP⁺ AVC and SAN CMs. High intensity mTagBFP2 cells with low or no mCherry signal was used to gate Atrial CMs. Ventricular CMs were gated for mCherry only positive cells as low mCherry signal was present in the atrium, and high mCherry in the ventricle (Supplementary Fig. 5a–d).

## Transcriptomic analysis

Total RNA was extracted from the sorted atrial CMs using the miR-Neasy micro kit (QIAGEN), combined with on-column DNase digestion (DNase-Free DNase Set, QIAGEN, 217084). Due to low RNA amount quality control steps were skipped. Sequencing was performed on a NextSeq2000 instrument (Illumina) using a P3 flowcell with $1 \times 72$ bp single end setup. Trimmomatic version 0.39 was employed to trim reads after a quality drop below a mean of Q15 in a window of 5 nucleotides and keeping only filtered reads longer than 15 nucleotides[110]. Reads were aligned versus Ensembl zebrafish genome version danRer11 (Ensembl release 104) with STAR 2.7.10a[111]. Alignments were filtered to remove: duplicates with Picard 3.0.0 (Picard: A set of tools (in Java) for working with next generation sequencing data in the BAM format), multi-mapping, ribosomal, or mitochondrial reads. Gene counts were established with featureCounts 2.0.4 by aggregating reads overlapping exons excluding those overlapping multiple genes[112]. The raw count matrix was normalized with DESeq2 version 1.36.0[113]. Contrasts were created with DESeq2 based on the raw count matrix. Genes were classified as significantly differentially expressed at average count > 5, multiple testing adjusted $p$-value < 0.05, and -0.585 <log2FC > 0.585. The Ensemble annotation was enriched with UniProt data (Activities at the Universal Protein Resource (UniProt)).

All downstream analyzes are based on the normalized gene count matrix. For KOBAS[114] gene set enrichment analysis, DEGs were split into up/down regulated genes. Significant gene set enrichment was defined by FDR and the top 10 gene sets or enriched pathways were plotted (dashed line: $p$-value = 0.05). For analysis of only the upregulated genes at 72 hpf, we used KEGG, a pathway enrichment analysis from Metascape[115].

## Statistics

All statistical analyzes and graphs, excluding the polarity plots and transcriptomic analyzes, were obtained using Graphpad Prism v9.3.1. Before choosing a statistical test, a Gaussian distribution was tested using the D'Agostino–Pearson omnibus and Shapiro-Wilk test for normality. All parametric data, which passed the normality tests, were analyzed using the two-way unpaired Student's t-test,

when two conditions were compared, or ordinary one-way ANOVA paired with Tukey's or Dunnett's multiple comparison test when multiple conditions were assessed. All non-parametric data that did not pass the normality test were analyzed using two-way Mann-Whitney's test when comparing two unpaired conditions, or two-tailed Wilcoxon test when comparing two paired conditions, and Kruskal-Wallis test paired with Dunn's multiple comparisons test when analyzing multiple conditions. The Student's t-test was paired with an F test to calculate the F-ratio in Supplementary Fig. 2. All error bars were calculated using mean with SD. The polarity graphs were plotted using R. For all graphs, the $p$-value cut-off used for significance is 0.05.

### Reporting summary
Further information on research design is available in the Nature Portfolio Reporting Summary linked to this article.

### Data availability
The bulk RNA-sequencing dataset reported in this paper was deposited in the Gene Expression Omnibus (GEO) database (accession: GSE249149). Source data are provided with this paper.

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

## Acknowledgements

We thank Virginie Lecaudey for comments and suggestions as a PhD thesis advisory committee member and for sharing reagents, Thomas Juan and Pinelopi Goumenaki for discussions and sharing reagents, Giulia Boezio for discussions and sharing protocols, David Sedmera and Sara Wickström for discussions and comments on the manuscript, Stefan Guenther for RNA-sequencing and analysis, Simon Perathoner for animal proposal support, and Radhan Ramadass for imaging support. This work was supported by funds from the Max Planck Society to D.Y.R.S.

## Author contributions

M.A. designed and performed experiments, analyzed the data, and wrote the manuscript. E.A. performed and analyzed experiments. A.G. and Y.X. helped with heart dissections, protocols, and discussions. K.K. performed FACS sorting. S.H. performed embryonic injections. C.K. helped with transcriptomic analysis. R.P. helped with reagents, transgenic fish lines, and project discussions. F.G. helped with heart dissections and supervised experiments. D.Y.R.S. helped to design experiments and analyze data, supervised the work and wrote the manuscript with input from all the authors.

## Funding

## Competing interests

The authors declare no competing interests.
