## [Peer Review File · Nature Communications]

Distinct mechanisms regulate ventricular and atrial chamber wall formationReviewer #1 (Remarks to the Author):

Albu et al. investigated the formation of the zebrafish atrium with longitudinal imaging. They observed that 25% of atrial cardiomyocytes elongated towards the heart's long axis and this elongation contributed to both cell intercalation and convergence thickening to create the internal network of the atrium. Further, they uncovered a role for Yap signaling in elongation of the atrial CMs with no evidence of Nrg/ErbB and Notch signaling. Overall, the manuscript is well-written, the microscopic imaging clearly demonstrates their findings, and they have uncovered new findings critical for ventricular and atrium development in the zebrafish.

Questions/Concerns:

1. In Figure 1B, why do the CM nuclei in the ventricle in the representative image have white and yellow CMs, instead of just yellow as observed with the atrial CMs?

Figure 1 F-H: Are the individual points plotted for every individual CM counted in each heart?

There is large deviation in CM shape index and angle.

Figure 1H'-H''': It might be beneficial to include a schematic, or add a cartoon representation to Figure 1A to depict how the atrial CM orientation was measured.

2. In Figure 2, the legend is missing information that the *cdh2:cdh2-EGFP* reporter, shown in cyan, is to label N-cadherin.

In figure 2c', the opaque 3D surfaces results showed elongating CM located above the neighboring CM in arrowed indicating sites, however, it looks like elongating CM located below the neighboring CM in the same position. Could the authors show different angles of the position to confirm the elongating CM located above the neighboring CM in Figure 2c.

In figure 2a'''-d''', which may determine the different localization of N-cadherin in elongating CM and neighboring CM?

3. In figure 3d-f', the author used mosaic labeling to track the same atrial CMs from 124 hpf to 14 dpf and observed that the elongated CMs present at 124 hpf form the internal muscle structures at 14 dpf. Do other CMs contribute to the internal muscle structure, did the authors observe the yellow inner muscle structure at 14dpf but no yellow signals in elongated CMs at 124hpf?

4. Figure 4H, 5E-F: discussion/explanation are needed considering there is much variation in the CM shape index, CM angle, but the p-value is considered significant.

5. The authors used *myh7: H2B-mScarle* to label atrial CMs. Is *myh6* reporter not available or does not recapitulate endogenous *myh6* expression in atrium?

6. The effects of K-975 and IWR-1 treatment on YAP signaling pathway need to be examined to confirm these drugs work in zebrafish. The discussion on the mechanism of YAP inhibiting atrial CM elongation will be appreciated.

7. Extended data Figure 4E-F: the labeling should be consistent, so either use Notch inhibitor in the image panel in 4E and y-axis of 4F, or say full drug name LY411575.

8. Extended data Figure 4J-M: accompanying graph/statistical analysis for DMSO vs BDM in J-K and *amhc +/?* vs *amhc* mutant in L-M should be added.

9. For all statistical tests conducted: what was the p-value cut-off used for significance? This should be included in the figure legends or in the Methods, Statistics subsection on pg. 24.

Reviewer #2 (Remarks to the Author):

Key results

The authors set out to investigate the origins of muscular ridges in the atrium during embryonic development, using zebrafish. They demonstrate that a subset of atrial CMs undergo changes in shape and radial intercalation prior to muscular ridge formation. This is accompanied by rearrangements in cell junctions. They describe protrusions in atrial CMs prior to cell elongation, and upon disruption of pathways required for the actin organisation that promotes cellular processes such as protrusion formation, demonstrate that cell elongation does not occur. They perform a thorough analysis of pathways that are important for ventricular wall maturation, and show that these are not important in atrial morphogenesis. They show that YAP activity is important for cell elongation in the atrium. The key novel findings from the paper are that: 1) changes in atrial morphology precede initiation of atrial ridge formation; 2) atrial cells display protrusions prior to cell shape changes; 3) atrial and ventricular walls have distinct mechanisms for building muscular networks during maturation.

Validity

In general the study is well-designed, well-executed, and well-reported. The findings are mostly well-supported by the data. Some conclusions could be better supported by an expanded analysis of their models (see data and methodology section for details).

Significance

While the process of ventricular trabeculation is relatively well-studied, maturation of the atrial wall is still very poorly understood. Defects in atrial morphology or function have significant physiological consequences, therefore this study makes an important contribution to a more overlooked aspect of heart development. Few studies have analysed later stages of atrial wall development, and without the mechanistic investigations delivered here.

Data and methodology

The data in the paper is beautifully presented, with high-quality 3D imaging of the live zebrafish heart. The study is well-informed by previous research in the field, making it extensive and thorough in interrogating the mechanisms that may regulate early aspects of atrial wall maturation. The methodology is detailed.

The authors use a variety of appropriate approaches to build their story. Novel transgenic lines have been generated to facilitate imaging the atrium (previously lacking in the field, therefore useful resources), and a wide variety of genetic tools for manipulating pathways or processes have been used. In most cases these are appropriate, and displayed with the relevant controls, although for one or two experiments (e.g. the *myl9* tools), the specificity of them for assaying e.g. protrusion formation, without performing the experiments that demonstrate that protrusion formation has been disrupted, renders some of the conclusions based around these experiments broader than those the authors draw.

The exploration of the relationship between the three phenomena the authors describe (protrusion formation, cell elongation, and then convergent thickening of the atrial wall) could be deeper. Much of the mechanistic analysis is performed using cell elongation as a readout, without looking at the impact of pathway disruption on either proliferation (to strengthen the link between the two), or in ridge formation (to strengthen the relationship between the latter). While the study does a lot of work to cross pathways off the list that are required for ventricular development, and identifies a pathway that regulates cell elongation in the atrium, the links between the whole process of atrial wall maturation remain a little inferred. This relationship is a significant early conclusion (and important contribution) of the manuscript. Questions thus remain whether, for example, in models where elongation is disrupted - does ridge formation fail to occur? In models where elongation is disrupted - did protrusion fail to occur earlier?

Analytical approach

Methodology for analysing the data presented seems appropriate, and the relevant controls have been included. Much of the data presented is quantified, and appropriate statistical tests and power have been used.

Suggested improvements

The authors conclude that 'cardiomyocyte elongation is an active process driven by membrane protrusion formation'. However, in their models that are used to manipulate protrusion formation, while they show impacts on cell elongation, they do not show that these processes have impacted protrusion formation. This is important since both IRSp53 and MYL9 will be regulating other processes in the cell (for example, IRSp53 regulates membrane-curvature-dependent processes, including protrusion formation, but also endosome dynamics, podosome formation, and polarised transport). The authors should strengthen their conclusions by showing that the IRSp53 and MYL9 dominant negative models reduce protrusion number or length in CMs prior to cell elongation at the same stage as shown in Figure 2c (76hpf), using the Akt1-Pest mosaic analysis as in Figure 4.

Similarly, the authors state that 'Both K-975 and IWR-1 treatments reduced atrial CM elongation (Fig. 5a-h)'. They also state that 'At the cellular level, atrial CM elongation is an active process driven by cytoskeletal changes (Fig. 4)'. In the latter, the authors are again making the specific link that cell protrusions are driving the cell elongations. How do these treatments affect protrusion? And does this affect convergent thickening? Interestingly, in Extended Figure 8b, both wild type embryos treated with K-975 and IWR-1 appear to have protrusions on their membranes - does this mean that protrusions form, but the elongation doesn't occur? Are they delayed? The authors could also strengthen this conclusion using the Akt-Pest1 mosaic analysis.

The authors show that Notch, Erbb2, and contractility are not required for cell elongation, and conclude that they are not required for atrial morphogenesis. Related to general comments about linking protrusion through elongation to convergent thickening - did they check that no radial intercalation is occurring in some/all of these models?

It is not clear from Figure 3b that the outer wall does contain a mixture of rounder and elongated cells - could the authors quantify this to strengthen their argument? Similarly, in Figure 3f - one of the yellow labelled cells looks quite rounded - is it then forming a ridge? Its initial morphology

looks very different from the other cell in the image. If the authors could quantify the 3D cell index for those longitudinal experiments and correlate the 3D index with location (muscle or outer wall) this would strengthen their conclusions.

How do the authors interpret the data that shows blocking YAP activity does result in reduced trabeculae? They state in the results that onset of trabeculation is not defective, but there is a reduction in the number of trabecular seeds. This could be addressed in the discussion.

If the outer myocardial wall also still contains elongated CMs, does this mean that elongation is required for ridge formation, but does not drive it? Why do some elongated cells drive ridge formation and not others? This could be addressed in the discussion.

Minor:

In Figure 2, atrial cardiomyocytes are labelled as having apical or basal adhesion. Do apical cardiomyocytes exhibit apicobasal polarity at this stage?

The authors state that 'high levels of YAP components' are seen at 48hpf and 72hpf, but what is high relative to?

It could be helpful to refer to the intercalation the authors describe more specifically as 'radial intercalation'.

Line 223/224 - there is a double 'visible'

Line 322 - extra 'k'

Clarity and context

The manuscript is well written, particularly given the number and complexity of zebrafish models used. The discussion is relatively brief, and could explore in more depth some questions raised by the data presented (for example, the authors note that while ridge atrial cardiomyocytes have elongated morphologies, outer wall cardiomyocytes contain a mixture of atrial and rounded - will these CMs in the outer wall eventually also form ridges, or do only a subset undergo radial intercalation?)

References

The manuscript cites relevant literature extensively.

Reviewer #3 (Remarks to the Author):

Much effort in understanding cardiac development has focused on ventricular development. The present study extends on this literature to focus on mechanisms of zebrafish atrial development. Here, the authors demonstrate that atrial development is accomplished by elongation of a subset of cardiomyocytes that go on to comprise the internal muscle structures of the zebrafish atrium. Through a series of pharmacological experiments, the authors conclude that atrial cardiomyocyte elongation is regulated, at least in part, by the Hippo pathway. This study contributes novel findings regarding the morphological events that occur in atrial cardiomyocytes during development. Larval zebrafish heart imaging is exceptionally well done,

and a variety of transgenic lines and pharmacological tools are employed throughout to determine mechanisms by which atrial elongation occurs. There are several key points that should be addressed primarily with regards to the mechanistic aspect of this study:

Major comments

Quantitative analysis should be provided to better support the claim that 'CM elongation and intercalation build the internal muscle structures in the zebrafish atrium' (re; figure 3).

It is unclear how bioinformatics analysis of RNAsequencing data lands on Hippo signaling as a lead candidate for subsequent investigation. As written, it seems that the investigators selected a handful of differentially expressed genes that happen to also be regulated by Hippo signaling, rather than employing an unbiased upstream regulator or KEGG pathway analysis to identify candidates. Is Hippo signaling a top significantly enriched pathway in differentially expressed gene set? While unbiased gene ontology analysis is shown, unbiased upstream regulator analysis, or similar, should be provided.

Much of the mechanistic insight in this manuscript depends on drug treatments inhibiting Yap/Wwtr1 activity. The Hippo pathway is so critical in organ development in general; off target effects of the drug or whole organism developmental defects could contribute to observed atrial cardiomyocyte morphology or cardiac function phenotypes observed in the present study. The discrepancy between YapDN and pharmacological experiments remains confusing and could be attributed to off target effects of the drug, or perhaps elements of the YapDN are not working as expected. Additional genetic experiments could reconcile these contradictory findings.

What is the effect of cardiomyocyte elongation on atrial function in adult zebrafish?

Minor comments:

Clarify description of Myh6/7 expression referred to on lines 67-68 as these differ across species and developmental timepoints.

Double check p-values in extended data Fig. 2 F,G as these are exactly the same value in both panels.

Raw reads in extended figure 6d is difficult to conceptualize without a reference.

Clarify cardiac function experiments in figure 9F. When are inhibitor given during development? What age is cardiac function assessed?

Line 322 see typo "k"

Reviewer #1

Albu et al. investigated the formation of the zebrafish atrium with longitudinal imaging. They observed that 25% of atrial cardiomyocytes elongated towards the heart's long axis and this elongation contributed to both cell intercalation and convergence thickening to create the internal network of the atrium. Further, they uncovered a role for Yap signaling in elongation of the atrial CMs with no evidence of Nrg/ErbB and Notch signaling. Overall, the manuscript is well-written, the microscopic imaging clearly demonstrates their findings, and they have uncovered new findings critical for ventricular and atrium development in the zebrafish.

We thank Reviewer #1 for their supportive comments.

Questions/Concerns:

1. In Figure 1B, why do the CM nuclei in the ventricle in the representative image have white and yellow CMs, instead of just yellow as observed with the atrial CMs?

CM reporter lines in zebrafish have a higher signal in the ventricle than in the atrium. Therefore, we adjusted the microscopy parameters to obtain clear imaging of the atrium, leading to signal saturation in the ventricle. We have now added a sentence in the methods section to clarify this point.

Figure 1 F-H: Are the individual points plotted for every individual CM counted in each heart? There is large deviation in CM shape index and angle.

Yes, individual points are plotted for Figures 1f and h, which results in a high variability caused by CM heterogeneity. In contrast, in Figure 1g the data points represent the percentage of elongating atrial CMs per heart. We have now added more information in all the relevant figure legends to clarify these points.

Figure 1H'-H''': It might be beneficial to include a schematic, or add a cartoon representation to Figure 1A to depict how the atrial CM orientation was measured.

We have now added a schematic regarding atrial CM angle determination in Figure 1b'.

2. In Figure 2, the legend is missing information that the *cdh2:cdh2-EGFP* reporter, shown in cyan, is to label N-cadherin.

We thank the reviewer for pointing out this oversight. We have now added the missing information.

In Figure 2c', the opaque 3D surfaces results showed elongating CM located above the neighboring CM in arrowed indicating sites, however, it looks like elongating CM located below the neighboring CM in the same position. Could the authors show different angles of the position to confirm the elongating CM located above the neighboring CM in Figure 2c.

We thank the reviewer for this suggestion. We have now added more information in the figure legend and results sections to clarify that the magenta surface marks the mTagBFP⁺ CM, which becomes located on the abluminal, or external, side of the atrium, i.e., on top of the mTagBFP⁻ CM shown with white surfaces, which itself becomes located on the luminal, or internal, side of the atrium.

In Figure 2a'''-d'''', which may determine the different localization of N-cadherin in elongating CM and neighboring CM?

We agree with the reviewer that investigating the mechanisms leading to the differential N-cadherin localization in atrial CMs is a worthwhile project for a follow up study.

3. In Figure 3d-f', the author used mosaic labeling to track the same atrial CMs from 124 hpf to 14 dpf and observed that the elongated CMs present at 124 hpf form the internal muscle structures at 14 dpf. Do other CMs contribute to the internal muscle structure, did the authors observe the yellow inner muscle structure at 14dpf but no yellow signals in elongated CMs at 124hpf?

We have now included additional data to address this point (Extended Data Figure 2). The current data suggest that all the inner muscle structures at 14 dpf originate from elongating CMs.

4. Figure 4H, 5E-F: discussion/explanation are needed considering there is much variation in the CM shape index, CM angle, but the p-value is considered significant.

The variability appears only in data from single CMs due to tissue heterogeneity. On the one hand, we observed a high angle variability prior to atrial CM elongation, meaning a stochastic CM orientation. But as atrial CMs elongate, they orient along the long axis of the heart, thereby reducing this variability. On the other hand, we observed that prior to atrial CM elongation, most atrial CM shape indices are closer to 1 indicating a rounder configuration. However, as atrial CMs elongate, a subset of CMs have a shape index closer to 0, but not all, thereby resulting in the observed increase in variability. This CM shape variability is thus indicative of the emerging tissue heterogeneity. We have added more information in all the relevant figure legends to clarify these points.

5. The authors used myh7: H2B-mScarlet to label atrial CMs. Is myh6 reporter not available or does not recapitulate endogenous myh6 expression in atrium?

We used *myl7:H2B-mScarlet* not a *myh7* driven reporter. *myl7* is expressed in all CMs including all atrial CMs, whereas *myh7* is expressed in ventricular CMs and only in some atrial CMs. Using Imaris we can segment out the atrial signal in 3D (as shown in Figure 1b', Movie 1) and quantify only the atrial signal even when using pan-cardiomyocyte markers. There is no published nuclear reporter line driven by the *myh6* promoter; and since *myh6* is also expressed in the AVC CMs and some ventricular CMs (Extended Data Figure 7), it would not improve the segmentation when compared with *myl7*.

6. The effects of K-975 and IWR-1 treatment on YAP signaling pathway need to be examined to confirm these drugs work in zebrafish. The discussion on the mechanism of YAP inhibiting atrial CM elongation will be appreciated.

Previous studies have used these inhibitors in zebrafish¹⁻³ and we have now referenced them in the manuscript. We have also now discussed the potential mechanism of Yap1 function in more detail.

7. Extended data Figure 4E-F: the labeling should be consistent, so either use Notch inhibitor in the image panel in 4E and y-axis of 4F, or say full drug name LY411575.

Noted and done.

8. Extended data Figure 4J-M: accompanying graph/statistical analysis for DMSO vs BDM in J-K and amhc +/- vs amhc mutant in L-M should be added.

Noted and done.

9. For all statistical tests conducted: what was the p-value cut-off used for significance? This should be included in the figure legends or in the Methods, Statistics subsection on pg. 24.

Noted and done. The p value cut off is 0.05.

Reviewer #2

Key results

The authors set out to investigate the origins of muscular ridges in the atrium during embryonic development, using zebrafish. They demonstrate that a subset of atrial CMs undergo changes in shape and radial intercalation prior to muscular ridge formation. This is accompanied by rearrangements in cell junctions. They describe protrusions in atrial CMs prior to cell elongation, and upon disruption of pathways required for the actin organisation that promotes cellular processes such as protrusion formation, demonstrate that cell elongation does not occur. They perform a thorough analysis of pathways that are important for ventricular wall maturation, and show that these are not important in atrial morphogenesis. They show that YAP activity is important for cell elongation in the atrium. The key novel findings from the paper are that: 1) changes in atrial morphology precede initiation of atrial ridge formation; 2) atrial cells display protrusions prior to cell shape changes; 3) atrial and ventricular walls have distinct mechanisms for building muscular networks during maturation.

Validity

In general the study is well-designed, well-executed, and well-reported. The findings are mostly well-supported by the data. Some conclusions could be better supported by an expanded analysis of their models (see data and methodology section for details).

Significance

While the process of ventricular trabeculation is relatively well-studied, maturation of the atrial wall is still very poorly understood. Defects in atrial morphology or function have significant physiological consequences, therefore this study makes an important contribution to a more overlooked aspect of heart development. Few studies have analysed later stages of atrial wall development, and without the mechanistic investigations delivered here.

Data and methodology

The data in the paper is beautifully presented, with high-quality 3D imaging of the live zebrafish heart. The study is well-informed by previous research in the field, making it extensive and thorough in interrogating the mechanisms that may regulate early aspects of atrial wall maturation. The methodology is detailed.

The authors use a variety of appropriate approaches to build their story. Novel transgenic lines have been generated to facilitate imaging the atrium (previously lacking in the field, therefore useful resources), and a wide variety of genetic tools for manipulating pathways or processes have been used. In most cases these are appropriate, and displayed with the relevant controls, although for one or two experiments (e.g. the *myl9* tools), the specificity of them for assaying e.g. protrusion formation, without performing the experiments that demonstrate that protrusion formation has been disrupted, renders some of the conclusions based around these experiments broader than those the authors draw.

The exploration of the relationship between the three phenomena the authors describe (protrusion formation, cell elongation, and then convergent thickening of the atrial wall) could be deeper. Much of the mechanistic analysis is performed using cell elongation as a readout, without looking at the impact of pathway disruption on either proliferation (to strengthen the link between the two), or in ridge formation (to strengthen the relationship between the latter). While the study does a lot of work to cross pathways off the list that are required for ventricular development, and identifies a pathway that regulates cell elongation in the atrium, the links between the whole process of atrial wall maturation remain a little inferred. This relationship is a significant early conclusion (and important contribution) of the manuscript. Questions thus remain whether, for example, in models where elongation is disrupted - does ridge formation fail to occur? In models where elongation is disrupted - did protrusion fail to occur earlier?

Analytical approach

Methodology for analysing the data presented seems appropriate, and the relevant controls have been included. Much of the data presented is quantified, and appropriate statistical tests and power have been used.

We thank Reviewer #2 for their supportive comments.

Suggested improvements

The authors conclude that ‘cardiomyocyte elongation is an active process driven by membrane protrusion formation’. However, in their models that are used to manipulate protrusion formation, while they show impacts on cell elongation, they do not show that these processes have impacted protrusion formation. This is important since both IRSp53 and MYL9 will be regulating other processes in the cell (for example, IRSp53 regulates membrane-curvature-dependent processes, including protrusion formation, but also endosome dynamics, podosome formation, and polarised transport). The authors should strengthen their conclusions by showing that the IRSp53 and MYL9 dominant negative models reduce protrusion number or length in CMs prior to cell elongation at the same stage as shown in Figure 2c (76hpf), using the Akt1-Pest mosaic analysis as in Figure 4.

We thank the reviewer for their constructive suggestions. We have now added quantifications in terms of the number of membrane protrusions at 76 hpf following IRSp53^{DN} or MYL9^{DN} overexpression (Extended Data Figure 5). To avoid possible side effects from reporter injections, we used the stable CM membrane reporter line to determine the number of membrane protrusions per atrial CM.

Through these experiments we observed that the number of membrane protrusions was only reduced in the IRSp53^{DN} overexpressing larvae. These data suggest that the loss of atrial CM elongation and the inability of CMs to orient when IRSp53^{DN} is overexpressed might be due to a reduction in the number of membrane protrusions. However, in the case of MYL9^{DN} overexpression it is likely that the change in contractility affects specifically the ability of CMs to produce large membrane protrusions and not the formation of smaller membrane protrusions. In summary, the data from the IRSp53^{DN} overexpressing larvae suggest that small protrusions present prior to CM elongation are required for this process; however, additional lines of evidence will be required to make a conclusive statement.

Similarly, the authors state that ‘Both K-975 and IWR-1 treatments reduced atrial CM elongation (Fig. 5a-h)’. They also state that ‘At the cellular level, atrial CM elongation is an active process driven by cytoskeletal changes (Figure 4)’. In the latter, the authors are again making the specific link that cell protrusions are driving the cell elongations. How do these treatments affect protrusion? And does this affect convergent thickening? Interestingly, in Extended Figure 8b, both wild type embryos treated with K-975 and IWR-1 appear to have protrusions on their membranes - does this mean that protrusions form, but the elongation doesn’t occur? Are they delayed? The authors could also strengthen this conclusion using the Akt-Pest1 mosaic analysis.

We thank the reviewer for these questions. We have now included additional quantifications in terms of the number of atrial CM membrane protrusions in larvae treated with DMSO, K-975, or IWR-1 from 72 to 100 hpf (Extended Data Figure 10h-k). Through these experiments we observed no significant change in the number of small membrane protrusions, suggesting that the phenotype caused by these inhibitor treatments is not a consequence of an effect on the formation of the small membrane protrusions, but rather on the formation of the larger membrane protrusions.

The authors show that Notch, Erbb2, and contractility are not required for cell elongation, and conclude that they are not required for atrial morphogenesis. Related to general comments about linking protrusion through elongation to convergent thickening - did they check that no radial intercalation is occurring in some/all of these models?

We thank the reviewer for this suggestion. We have now added 3D surface renderings of the inner atrial myocardium from all these models in Extended Data Figure 6, and found no evidence for a loss of convergent thickening (as observed by the presence of inner myocardial ridges in all these models). Conversely, we observed a loss of inner atrial myocardial ridges following IWR-1 and K-975 treatments (Extended Data Figure 11h-j).

It is not clear from Figure 3b that the outer wall does contain a mixture of rounder and elongated cells - could the authors quantify this to strengthen their argument? Similarly, in Figure 3f - one of the yellow labelled cells looks quite rounded - is it then forming a ridge? Its initial morphology looks very different from the other cell in the image. If the authors could quantify the 3D cell index for those longitudinal experiments and correlate the 3D index with location (muscle or outer wall) this would strengthen their conclusions.

We thank the reviewer for their feedback. We have now added quantifications for this experiment as well (Extended Data Figure 2). These quantifications reveal that the outer layer of the 14 dpf atrial myocardium comprises both elongating and round CMs. Furthermore, through quantifying the 3D shape index of the mTagBFP⁺ atrial CMs, we observed that all elongating atrial CMs at 124 hpf retain their elongated shape but that only approximately 50% of the elongating CMs form the inner muscle structures. In the case of round atrial CMs at 124 hpf, we observed that a subset of them elongate and some of these then go on to form inner muscle structures by 14 dpf; however, none of the round atrial CMs at 14 dpf are part of the inner muscle structures. Altogether, these quantifications solidify the link between atrial CM elongation, intercalation, and subsequent inner muscle structure formation.

How do the authors interpret the data that shows blocking YAP activity does result in reduced trabeculae? They state in the results that onset of trabeculation is not defective, but there is a reduction in the number of trabecular seeds. This could be addressed in the discussion.

We have now discussed this point.

If the outer myocardial wall also still contains elongated CMs, does this mean that elongation is required for ridge formation, but does not drive it? Why do some elongated cells drive ridge formation and not others? This could be addressed in the discussion.

We thank the reviewer for these interesting questions which we have now addressed in the discussion.

Minor:

In Figure 2, atrial cardiomyocytes are labelled as having apical or basal adhesion. Do apical cardiomyocytes exhibit apicobasal polarity at this stage?

Yes, atrial CMs exhibit apicobasal polarity at this stage (see new Extended Data Fig. 1a-b’)

The authors state that ‘high levels of YAP components’ are seen at 48hpf and 72hpf, but what is high relative to?

We thank the reviewer for this question. We have now removed the word “high” from the manuscript, since we are indeed not comparing the expression of these genes to another parameter. Nevertheless, the data still suggest that these Hippo pathway genes are expressed in atrial CMs.

It could be helpful to refer to the intercalation the authors describe more specifically as ‘radial intercalation’.

Radial intercalation usually occurs in a multilayered tissue and results in thinning/spreading⁴. We observe the opposite effect whereby there is a monolayer that forms regions of multilayering through intercalation, which has previously been referred to as convergent thickening⁵. Therefore, we refer to this atrial CM intercalation process as convergent thickening.

Line 223/224 - there is a double ‘visible’

Thank you; corrected.

Line 322 - extra ‘k’

Thank you; corrected.

Clarity and context

The manuscript is well written, particularly given the number and complexity of zebrafish models used. The discussion is relatively brief, and could explore in more depth some questions raised by the data presented (for example, the authors note that while ridge atrial cardiomyocytes have elongated morphologies, outer wall cardiomyocytes contain a mixture of atrial and rounded - will these CMs in the outer wall eventually also form ridges, or do only a subset undergo radial intercalation?)

We thank the reviewer for these interesting questions. We have expanded the discussion accordingly.

References

The manuscript cites relevant literature extensively.

We thank the reviewer for their positive comment.

Reviewer #3 (Remarks to the Author):

Much effort in understanding cardiac development has focused on ventricular development. The present study extends on this literature to focus on mechanisms of zebrafish atrial development. Here, the authors demonstrate that atrial development is accomplished by elongation of a subset of cardiomyocytes that go on to comprise the internal muscle structures of the zebrafish atrium. Through a series of pharmacological experiments, the authors conclude that atrial cardiomyocyte elongation is regulated, at least in part, by the Hippo pathway. This study contributes novel findings regarding the morphological events that occur in atrial cardiomyocytes during development. Larval zebrafish heart imaging is exceptionally well done, and a variety of transgenic lines and pharmacological tools are employed throughout to determine mechanisms by which atrial elongation occurs. There are several key points that should be addressed primarily with regards to the mechanistic aspect of this study:

We thank Reviewer #3 for their supportive comments.

Major comments

Quantitative analysis should be provided to better support the claim that ‘CM elongation and intercalation build the internal muscle structures in the zebrafish atrium’ (re; figure 3).

We thank the reviewer for their suggestion. We have now quantified the data from this experiment and found that indeed only elongating atrial CMs form the internal muscle structures in the zebrafish atrium (Extended Data Figure 2).

It is unclear how bioinformatics analysis of RNAsequencing data lands on Hippo signaling as a lead candidate for subsequent investigation. As written, it seems that the investigators selected a handful of differentially expressed genes that happen to also be regulated by Hippo signaling, rather than employing an unbiased upstream regulator or KEGG pathway analysis to identify candidates. Is Hippo signaling a top significantly enriched pathway in differentially expressed gene set? While unbiased gene ontology analysis is shown, unbiased upstream regulator analysis, or similar, should be provided. We did employ unbiased gene set overrepresentation analyses based on DEGs (via kobas) and found proliferation genes to be significantly upregulated (Extended Data Figure 8a). We thus sought to test pathways known to regulate cell proliferation. Since a previous study had reported the Hippo pathway to be a potential regulator of atrial CM number⁶, we decided to take a closer look at it. To answer the reviewer’s suggestion, we also carried out an unbiased upstream regulator analysis (Figure R1).

Transcription factor binding site enrichment analysis

Figure R1: Transcription factor binding site (TFBS) enrichment analysis was performed with Pscan⁷. Reference TFBS position weight matrices CORE_vertbrates_non-redundant_pfms were extracted from JASPAR on 20221124⁸. Overrepresented TFBS were identified based on the promoter nucleotide sequence (450-TSS-50) of protein coding genes using DEGs as foreground list and all genes as background. The resulting heatmap shows TFs that were significantly enriched with uncorrected p-value < 0.05 in at least one foreground list (yellow = overrepresented).

Much of the mechanistic insight in this manuscript depends on drug treatments inhibiting Yap/Wwtr1 activity. The Hippo pathway is so critical in organ development in general; off target effects of the drug or whole organism developmental defects could contribute to observed atrial cardiomyocyte morphology or cardiac function phenotypes observed in the present study. The discrepancy between YapDN and pharmacological experiments remains confusing and could be attributed to off target effects of the drug, or perhaps elements of the YapDN are not working as expected. Additional genetic experiments could reconcile these contradictory findings.

We fully agree with the reviewer which is why we stated in the discussion that more sophisticated tools are required to draw conclusions regarding a role for Yap1 in atrial CM elongation.

What is the effect of cardiomyocyte elongation on atrial function in adult zebrafish?

We thank the reviewer for their interesting question. While we acknowledge that it is an important point, it will require new genetic tools to address.

Minor comments:

Clarify description of Myh6/7 expression referred to on lines 67-68 as these differ across species and developmental timepoints.

Noted and done.

Double check p-values in extended data Fig. 2 F,G as these are exactly the same value in both panels.

Noted and done. The p values are correct.

Raw reads in extended figure 6d is difficult to conceptualize without a reference.

Noted and done.

Clarify cardiac function experiments in figure 9F. When are inhibitor given during development? What age is cardiac function assessed?

Noted and done.

Line 322 see typo “k”

Thank you; corrected.

References

1. Nicenboim, J. *et al.* Lymphatic vessels arise from specialized angioblasts within a venous niche. *Nature* **522**, 56–61 (2015).
2. Moon, J. & Amatruda, J. F. Biochemical Analysis of Tankyrase Activity in Zebrafish In Vitro and In Vivo. in *Wnt Signaling: Methods and Protocols* (eds. Barrett, Q. & Lum, L.) 95–100 (Springer, New York, NY, 2016). doi:10.1007/978-1-4939-6393-5_10.
3. Sturtzel, C. *et al.* Refined high-content imaging-based phenotypic drug screening in zebrafish xenografts. *npj Precis. Onc.* **7**, 1–16 (2023).
4. Wallingford, J. B., Fraser, S. E. & Harland, R. M. Convergent Extension: The Molecular Control of Polarized Cell Movement during Embryonic Development. *Developmental Cell* **2**, 695–706 (2002).
5. Keller, R. & Danilchik, M. Regional expression, pattern and timing of convergence and extension during gastrulation of *Xenopus laevis*. *Development* **103**, 193–209 (1988).
6. Fukui, H. *et al.* Hippo signaling determines the number of venous pole cells that originate from the anterior lateral plate mesoderm in zebrafish. *eLife* **7**, e29106 (2018).
7. Zambelli, F., Pesole, G. & Pavesi, G. Pscan: finding over-represented transcription factor binding site motifs in sequences from co-regulated or co-expressed genes. *Nucleic Acids Research* **37**, W247–W252 (2009).
8. Castro-Mondragon, J. A. *et al.* JASPAR 2022: the 9th release of the open-access database of transcription factor binding profiles. *Nucleic Acids Research* **50**, D165–D173 (2022).

Reviewer #1 (Remarks to the Author):

I am satisfied with the revisions and have no more comments.

Reviewer #2 (Remarks to the Author):

I thank the authors for their detailed and considered response to my review. The new analyses and quantifications strengthen the conclusions made in the manuscript, highlight areas for future consideration, and additions in the discussion are helpful for readers to contextualise the results. I have no further suggested revisions, and would recommend the manuscript is ready for publication.

Key results (reproduced from original review)

The authors set out to investigate the origins of muscular ridges in the atrium during embryonic development, using zebrafish. They demonstrate that a subset of atrial CMs undergo changes in shape and radial intercalation prior to muscular ridge formation. This is accompanied by rearrangements in cell junctions. They describe protrusions in atrial CMs prior to cell elongation, and upon disruption of pathways required for the actin organisation that promotes cellular processes such as protrusion formation, demonstrate that cell elongation does not occur. They perform a thorough analysis of pathways that are important for ventricular wall maturation, and show that these are not important in atrial morphogenesis. They show that YAP activity is important for cell elongation in the atrium. The key novel findings from the paper are that: 1) changes in atrial morphology precede initiation of atrial ridge formation; 2) atrial cells display protrusions prior to cell shape changes; 3) atrial and ventricular walls have distinct mechanisms for building muscular networks during maturation.

Reviewer #3 (Remarks to the Author):

The authors have sufficiently addressed all comments.